# Preference learning along multiple criteria:
# A game-theoretic perspective

**Kush Bhatia**
EECS, UC Berkeley
kush@cs.berkeley.edu

**Ashwin Pananjady**
Simons Institute, UC Berkeley
ashwinpm@berkeley.edu

**Peter L. Bartlett**
EECS and Statistics, UC Berkeley
peter@berkeley.edu

**Anca D. Dragan**
EECS, UC Berkeley
anca@berkeley.edu

**Martin J. Wainwright**
EECS and Statistics, UC Berkeley
wainwrig@berkeley.edu

## Abstract

The literature on ranking from ordinal data is vast, and there are several ways to aggregate overall preferences from pairwise comparisons between objects. In particular, it is well-known that any Nash equilibrium of the zero-sum game induced by the preference matrix defines a natural solution concept (winning distribution over objects) known as a von Neumann winner. Many real-world problems, however, are inevitably multi-criteria, with different pairwise preferences governing the different criteria. In this work, we generalize the notion of a von Neumann winner to the multi-criteria setting by taking inspiration from Blackwell's approachability. Our framework allows for non-linear aggregation of preferences across criteria, and generalizes the linearization-based approach from multi-objective optimization.

From a theoretical standpoint, we show that the Blackwell winner of a multi-criteria problem instance can be computed as the solution to a convex optimization problem. Furthermore, given random samples of pairwise comparisons, we show that a simple, "plug-in" estimator achieves (near-)optimal minimax sample complexity. Finally, we showcase the practical utility of our framework in a user study on autonomous driving, where we find that the Blackwell winner outperforms the von Neumann winner for the overall preferences.

## 1   Introduction

Economists, social scientists, engineers, and computer scientists have long studied models for human preferences, under the broad umbrella of social choice theory [10, 7]. Learning from human preferences has found applications in interactive robotics for learning reward functions [45, 39], in medical domains for personalizing assistive devices [59, 9], and in recommender systems for optimizing search engines [15, 28]. The recent focus on safety in AI has popularized human-in-the-loop learning methods that use human preferences in order to promote value alignment [16, 46, 6].

The most popular form of preference elicitation is to make pairwise comparisons [51, 13, 33]. Eliciting such feedback involves showing users a pair of objects and asking them a query: Do you prefer object A or object B? Depending on the application, an object could correspond to a product in a search query, or a policy or reward function in reinforcement learning. A vast body of classical work dating back to Condorcet and Borda [17, 12] has focused on defining and producing a "winning" object from the result of a set of pairwise comparisons.

In relatively recent work, Dudik et al. [22] proposed the concept of a von Neumann winner, corresponding to a distribution over objects that beats or ties every other object in the collection. They showed that under an expected utility assumption, such a randomized winner always exists and

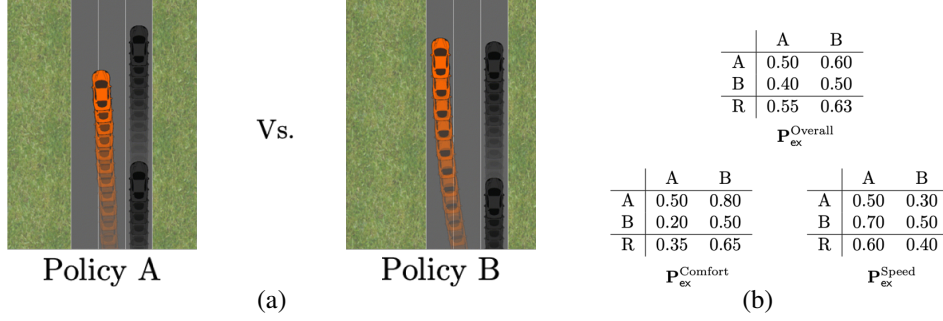

Vs.

Policy A          Policy B

(a)

|   | A | B |
|---|---|---|
| A | 0.50 | 0.60 |
| B | 0.40 | 0.50 |
| R | 0.55 | 0.63 |

$\mathbf{P}_{\text{ex}}^{\text{Overall}}$

|   | A | B |
|---|---|---|
| A | 0.50 | 0.80 |
| B | 0.20 | 0.50 |
| R | 0.35 | 0.65 |

$\mathbf{P}_{\text{ex}}^{\text{Comfort}}$

|   | A | B |
|---|---|---|
| A | 0.50 | 0.30 |
| B | 0.70 | 0.50 |
| R | 0.60 | 0.40 |

$\mathbf{P}_{\text{ex}}^{\text{Speed}}$

(b)

**Figure 1.** (a) Policy A focuses on optimizing comfort and policy B on speed, and these are compared pairwise in different environments. (b) Preference matrices, where entry $(i, j)$ of the matrix contains the proportion of comparisons between the pair $(i, j)$ that are won by object $i$. (The diagonals are set to half by convention). The overall pairwise comparisons are given by the matrix $\mathbf{P}_{\text{ex}}^{\text{Overall}}$, and preferences along each of the criteria by matrices $\mathbf{P}_{\text{ex}}^{\text{Comfort}}$ and $\mathbf{P}_{\text{ex}}^{\text{Speed}}$ (the numbers here are illustrative of our user-study in Section 4). Policy R is a randomized policy $1/2$ A $+ 1/2$ B. While the preference matrices satisfy the linearity assumption individually along speed and comfort, the assumption is violated overall, wherein R is preferred over both A and B.

overcomes limitations of existing winning concepts—the Condorcet winner does not always exist, while the Borda winner fails an independence of clones test [47]. However, the assumption of expected utility relies on a strong hypothesis about how humans evaluate distributions over objects: it posits that the probability with which any distribution over objects $\pi$ beats an object is linear in $\pi$.

**Consequences of assuming linearity:** In order to better appreciate these consequences, consider as an example the task of deciding between two policies (say A and B) to deploy in an autonomous vehicle. Suppose that these policies have been obtained by optimizing two different objectives, with policy A optimized for comfort and policy B optimized for speed. Figure 1(a) shows a snapshot of these two policies. When compared overall, 60% of the people preferred Policy A over B – making it the von Neumann winner. The linearity assumption then posits that a randomized policy that mixes between A and B can *never* be better than both A and B; but we see that the Policy R = $1/2$ A + $1/2$ B is actually preferred by a majority over both A and B! Why is the linearity assumption violated here?

One possible explanation for such a violation is that the comparison problem is actually *multi-criteria* in nature. If we look at the preferences for the criterion speed and comfort individually in Figure 1(b), we see that Policy A does quite poorly on the speed axis while B lags behind in comfort. In contrast, Policy R does acceptably well along both the criteria and hence is preferred overall to both Policies A and B. It is indeed impossible to come to this conclusion by only observing the overall comparisons. This observation forms the basis of our main proposal: decompose the single overall comparison and ask humans to provide preferences along *simpler* criteria. This decomposition of the comparison task allows us to place structural assumptions on comparisons along each criterion. For instance, we may now posit the linearity assumption along each criterion separately rather than on the overall comparison task. In addition to allowing for simplified assumptions, breaking up the task into such simpler comparisons allows us to obtain richer and more accurate feedback as compared to the single overall comparison. Indeed, such a motivation for eliciting simpler feedback from humans finds its roots in the the study of cognitive biases in decision making, which suggests that the human mind resorts to simple heuristics when faced with a complicated questions [53].

**Contributions:** In this paper, we formalize these insights and propose a new framework for preference learning when pairwise comparisons are available along multiple, possibly conflicting, criteria. As shown by our example in Figure 1, a single distribution which is the von Neumann winner along every criteria might not exist. To counter this, we formulate the problem of finding the "best" randomized policy by drawing on tools from the literature on vector valued pay-offs in game theory. Specifically, we take inspiration from Blackwell's approachability [11] and introduce the notion of a Blackwell winner. This solution concept strictly generalizes the concept of a von Neumann winner, and recovers the latter when there is only a single criterion present. Section 2 describes this framework in detail, and Section 3 collects our statistical and computational guarantees for learning the Blackwell winner from data. Section 4 describes a user study with an autonomous driving environment, in which we ask human subjects to compare self-driving policies along multiple

criteria such as safety, aggressiveness, and conservativeness. Our experiment demonstrates that the Blackwell winner is able to better trade off utility along these criteria and produces randomized policies that outperform the von Neumann winner for the overall preferences.

**Related work.** Most closely related to our work is the field of computational social choice, which has focused on defining notions of winners from overall pairwise comparisons (see the survey [37] for a review). Amongst them, three deterministic notions of a winner—the Condorcet [17], Borda [12], and Copeland [18] winners—have been widely studied. In addition, Dudik et al. [22] recently introduced the notion of a (randomized) von Neumann winner. Starting with the work of Yue et al. [57], there have been several research papers studying an online version of preference learning, called the Dueling Bandits problem. Algorithms have been proposed to compete with Condorcet [60, 62, 4], Copeland [61, 56], Borda [30] and von Neumann [22] winners.

The theoretical foundations of decision making based on multiple criteria have been widely studied within the operations research community . This sub-field—called multiple-criteria decision analysis—has focused largely on scoring, classification, and sorting based on multiple-criteria feedback. See the surveys [44, 63] for thorough overviews of existing methods and their associated guarantees. The problem of eliciting the user's relative weighting of the various criteria has also been considered [20]. However, relatively less attention has been paid to the study of randomized decisions and statistical inference, both of which form the focus of our work. From an applied perspective, the combination of multi-criteria assessments has received attention in disparate fields such as psychometrics [40, 35], healthcare [50], and recidivism prediction [55]. In many of these cases, a variety of approaches—both linear and non-linear—have been empirically evaluated [19]. Justification for non-linear aggregation of scores along the criteria has a long history in psychology and the behavioral sciences [27, 24, 54].

In the game theory literature, Blackwell [11] introduced the notion of approachability as a generalization of a zero-sum game with vector-valued payoffs (for a detailed discussion see Appendix A). Blackwell's approachability and its connections with no-regret learning and calibrated forecasting have been extensively studied [1, 42, 34]. These connections have enabled applications of Blackwell's results to problems ranging from constrained reinforcement learning [36] to uncertainty estimation for question-answering tasks [31]. In contrast, our framework for preference learning along multiple criteria deals with a single shot game and uses the idea of the target set to define the concept of a Blackwell winner. Another body of literature related to our work studies Nash equilibria in games with perturbed payoffs, under both robust [3, 32] and uncertain (or Bayesian) [25] formulations (see the recent survey by Perchet [43]). Perturbation theory for Nash equilibria has been derived in these contexts, and it is well-known that the Nash equilibrium is not (at least in general) stable to perturbations of the payoff matrix. On the other hand, the results of [22] consider Nash equilibria of perturbed, symmetric, zero-sum games, but show that the *payoff* of the perturbed Nash equilibrium is indeed stable. Our work provides a similar characterization for the multi-criteria setting.

## 2 Framework for preference learning along multiple criteria

We now set up our framework for preference learning along multiple criteria. We consider a collection of $d$ objects over which comparisons can be elicited along $k$ different criteria. We index the objects by the set $[d] := \{1, \ldots, d\}$ and the criteria by the set $[k]$.

### 2.1 Probabilistic model for comparisons

Since human responses to comparison queries are typically noisy, we model the pairwise preferences as random variables drawn from an underlying population distribution. In particular, the result of a comparison between a pair of objects $(i_1, i_2)$ along criterion $j$ is modeled as a draw from a Bernoulli distribution, with $p(i_1, i_2; j) = \mathbb{P}(i_1 \succeq i_2 \text{ along criterion } j)$. By symmetry, we must have

$$p(i_2, i_1; j) = 1 - p(i_1, i_2; j) \text{ for each triple } i_1 \in [d], \ i_2 \in [d], \ \text{and } j \in [k]. \tag{1}$$

We let $\pi_1, \pi_2 \in \Delta_d$ represent[1] two distributions over the $d$ objects. With a slight abuse of notation, let $p(\pi_1, \pi_2; j)$ denote the probability with which an object drawn from distribution $\pi_1$ beats an object

drawn from distribution $\pi_2$ along criterion $j$. We assume for each individual criterion $j$ that the probability $p(\pi_1, \pi_2; j)$ is linear in the distributions $\pi_1$ and $\pi_2$, i.e. that it satisfies the relation

$$p(\pi_1, \pi_2; j) := \mathbb{E}_{i_1 \sim \pi_1, i_2 \sim \pi_2} \left[ p(i_1, i_2; j) \right]. \tag{2}$$

Equation (2) encodes the per-criterion linearity assumption highlighted in Section 1. We collect the probabilities $\{p(i_1, i_2; j)\}$ into a *preference tensor* $\mathbf{P} \in [0, 1]^{d \times d \times k}$ and denote by $\mathcal{P}_{d,k}$ the set of all preference tensors that satisfy the symmetry condition (1). Specifically, we have

$$\mathcal{P}_{d,k} = \{\mathbf{P} \in [0, 1]^{d \times d \times k} \mid \mathbf{P}(i_1, i_2; j) = 1 - \mathbf{P}(i_2, i_1; j) \text{ for all } (i_1, i_2, j)\}. \tag{3}$$

Let $\mathbf{P}^j$ denote the $d \times d$ matrix corresponding to the comparisons along criterion $j$, so that $p(\pi_1, \pi_2; j) = \pi_1^\top \mathbf{P}^j \pi_2$. Also note that a comparison between a pair of objects $(i_1, i_2)$ induces a *score vector* containing $k$ such probabilities. Denote this vector by $\mathbf{P}(i_1, i_2) \in [0, 1]^k$, whose $j$-th entry is given by $p(i_1, i_2; j)$. Denote by $\mathbf{P}(\pi_1, \pi_2)$ the score vector for a pair of distribution $(\pi_1, \pi_2)$.

In the single criterion case when $k = 1$, each comparison between a pair of objects is along an *overall* criterion. We let $\mathbf{P}_{\mathsf{ov}} \in [0, 1]^{d \times d}$ represent such an overall comparison matrix. As mentioned in Section 1, most preference learning problems are multi-objective in nature, and the overall preference matrix $\mathbf{P}_{\mathsf{ov}}$ is derived as a non-linear combination of per-criterion preference matrices $\{\mathbf{P}^j\}_{j=1}^k$. Therefore, even when the linearity assumption (2) holds across each criterion, it might not hold for the *overall* preference $\mathbf{P}_{\mathsf{ov}}$. In contrast, when the matrices $\mathbf{P}^j$ are aggregated linearly to obtain the overall matrix $\mathbf{P}_{\mathsf{ov}}$, we recover the assumptions of Dudik et al. [22].

## 2.2 Blackwell winner

Given our probabilistic model for pairwise comparisons, we now describe our notion of a Blackwell winner. When defining a winning distribution for the multi-criteria case, it would be ideal to find a distribution $\pi^*$ that is a von Neumman winner along *each* of the criteria separately. However, as shown in our example from Figure 1, such a distribution need not exist. We thus need a generalization of the von Neumann winner that explicitly accounts for conflicts between the criteria.

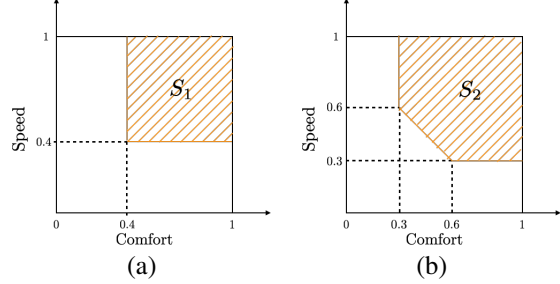

(a)          (b)

**Figure 2.** Two target sets $S_1$ and $S_2$ for our example from Figure 1 that capture trade-offs between comfort and speed. Set $S_1$ requires feasible score vectors to satisfy 40% of the population along both comfort and speed. Set $S_2$ requires both scores to be greater than 0.3 but with a linear trade-off: the combined score must be at least 0.9.

Blackwell [11] asked a related question for the theory of zero-sum games: how can one generalize von Neumann's minimax theorem to vector-valued games? He proposed the notion of a *target set*: a set of acceptable payoff vectors that the first player in a zero-sum game seeks to attain. Within this context, Blackwell proposed the notion of approachability, i.e. how the player might obtain payoffs in a repeated game that are close to the target set on average. We take inspiration from these ideas to define a solution concept for the multi-criteria preference problem. Our notion of a winner also relies on a *target set*, which we denote by $S \subset [0, 1]^k$, and which in our setting contains *score vectors*. This set provides a way to combine different criteria by specifying combinations of preference scores that are acceptable. Figure 2 provides an example of two such sets.

Observe that for our preference learning problem, the target set $S$ is by definition monotonic with respect to the orthant ordering, that is, if $z_1 \geq z_2$ coordinate-wise, then $z_2 \in S$ implies $z_1 \in S$. Our goal is to then produce a distribution $\pi^*$ that can achieve a target score vector for any distribution with which it is compared—that is $\mathbf{P}(\pi^*, \pi) \in S$ for all $\pi \in \Delta_d$. When such a distribution $\pi^*$ exists, we say that the problem instance $(\mathbf{P}, S)$ is *achievable*. On the other hand, it is clear that there are problem instances $(\mathbf{P}, S)$ that are not achievable. While Blackwell's workaround was to move to the setting of repeated games, preference aggregation is usually a one-shot problem. Consequently, our relaxation instead introduces the notion of a *worst-case distance* to the target set. In particular, we measure the distance between any pair of score vectors $u, v \in [0, 1]^k$ as $\rho(u, v) = \|u - v\|$ for some norm $\|\cdot\|$. Using the shorthand $\rho(u, S) := \inf_{v \in S} \|u - v\|$, the *Blackwell winner* $\pi^*$ for an instance $(\mathbf{P}, S, \|\cdot\|)$ is now defined as the one which minimizes the maximum distance to the set $S$, i.e.,

$$\pi(\mathbf{P}, S, \|\cdot\|) \in \underset{\pi \in \Delta_d}{\operatorname{argmin}}[v(\pi; \mathbf{P}, S, \|\cdot\|)], \quad \text{where} \quad v(\pi; \mathbf{P}, S, \|\cdot\|) := \max_{\pi' \in \Delta_d} \rho(\mathbf{P}(\pi, \pi'), S). \tag{4}$$

Observe that equation (4) has an interpretation as a zero-sum game, where the objective of the minimizing player is to make the score vector $\mathbf{P}(\pi, \pi')$ as close as possible to the target set $S$.

We now look at commonly studied frameworks for single criterion preference aggregation and multi-objective optimization and show how these can be naturally derived from our framework.

**Example: Preference learning along a single criterion.** A particular special case of our framework is when we have a single criterion ($k = 1$) and the preferences are given by a matrix $\mathbf{P}_{\mathsf{ov}}$. The score $\mathbf{P}_{\mathsf{ov}}(i_1, i_2)$ is a scalar representing the probability with which object $i_1$ beats object $i_2$ in an overall comparison. As a consequence of the von Neumann minimax theorem, we have

$$\max_{\pi_1 \in \Delta_d} \min_{\pi_2 \in \Delta_d} \mathbf{P}_{\mathsf{ov}}(\pi_1, \pi_2) = \min_{\pi_2 \in \Delta_d} \max_{\pi_1 \in \Delta_d} \mathbf{P}_{\mathsf{ov}}(\pi_1, \pi_2) = \frac{1}{2}, \tag{5}$$

with any maximizer above called the von Neumann winner [22]. Thus, for *any* preference matrix $\mathbf{P}_{\mathsf{ov}}$, a von Neumann winner is preferred to any other object with probability at least $\frac{1}{2}$.

Let us show how this uni-criterion formulation can be derived as a special case of our framework. Consider the target set $S = [\frac{1}{2}, 1]$ and choose the distance function $\rho(a, b) = |a - b|$. By equation (5), the target set $S = [\frac{1}{2}, 1]$ is achievable *for all* preference matrices $\mathbf{P}_{\mathsf{ov}}$, and so the von Neumann winner and the Blackwell winner $\pi(\mathbf{P}_{\mathsf{ov}}, [\frac{1}{2}, 1], |\cdot|)$ coincide. ♣

**Example: Weighted combinations of a multi-criterion problem.** One of the common approaches used in multi-objective optimization to reduce a multi-dimensional problem to a uni-dimensional counterpart is by introducing a weighted combinations of objectives. Formally, consider a weight vector $w \in \Delta_k$ and the corresponding preference matrix $\mathbf{P}(w) := \sum_{j \in [k]} w_j \mathbf{P}^j$ obtained by combining the preference matrices along the different criteria. A winning distribution can then be obtained by solving for the von Neumann winner of $\mathbf{P}(w)$ given by $\pi(\mathbf{P}(w), [\frac{1}{2}, 1], |\cdot|)$. The following proposition establishes that such an approach is a particular special case of our framework.

**Proposition 1.** *(a) For every weight vector $w \in \Delta_k$, there exists a target set $S_w \in [0, 1]^k$ such that for any norm $\|\cdot\|$, we have*

$$\pi(\mathbf{P}, S_w, \|\cdot\|) = \pi(\mathbf{P}(w), [1/2, 1], |\cdot|) \quad \text{for all} \quad \mathbf{P} \in \mathcal{P}_{d,k}.$$

*(b) Conversely, there exists a set $S$ and a preference tensor $\mathbf{P}$ with a* unique *Blackwell winner $\pi^*$ such that for all $w \in \Delta_k$, exactly one of the following is true:*

$$\pi(\mathbf{P}(w), [1/2, 1], |\cdot|) \neq \pi^* \quad \text{or} \quad \underset{\pi \in \Delta_d}{\operatorname{argmax}} \min_{i \in [d]} \mathbf{P}(\pi, i) = \Delta_d .$$

Thus, while the Blackwell winner is always able to recover any linear combination of criteria, the converse is not true. Specifically, part (b) of the proposition shows that for a choice of preference tensor $\mathbf{P}$ and target set $S$, either the von Neumann winner for $\mathbf{P}(w)$ is not equal to the Blackwell winner, or it degenerates to the entire simplex $\Delta_d$ and is thus uninformative. Consequently, our framework is strictly more general that weighting the individual criteria. ♣

## 3 Statistical guarantees and computational approaches

In this section, we provide theoretical results on computing the Blackwell winner from samples of pairwise comparisons along the various criteria.

**Observation model and evaluation metrics.** We operate in the natural passive observation model, where a sample consists of a comparison between two randomly chosen objects along a randomly chosen criterion. Specifically, we assume access to an oracle that when queried with a tuple $\eta = (i_1, i_2, j)$ comprising a pair of objects $(i_1, i_2)$ and a criterion $j$, returns a comparison $y(\eta) \sim \mathsf{Ber}(p(i_1, i_2; j))$. Each query to the oracle constitutes one sample. In the passive sampling model, the tuple of objects and criterion is sampled uniformly, with replacement, that is $(i_1, i_2) \sim \mathsf{Unif}\{\binom{[d]}{2}\}$ and $j \sim \mathsf{Unif}\{[k]\}$ where $\mathsf{Unif}\{A\}$ denotes the uniform distribution over the elements of a set $A$. Given access to samples $\{y_1(\eta_1), \ldots, y_n(\eta_n)\}$ from this observation model, we define the empirical preference tensor (specifically the upper triangular part)

$$\widehat{\mathbf{P}}_n(i_1, i_2, j) := \frac{\sum_{\ell=1}^{n} y_\ell(\eta_\ell) \mathbb{I}[\eta_\ell = (i_1, i_2, j)]}{1 \vee \sum_\ell \mathbb{I}[\eta_\ell = (i_1, i_2, j)]} \quad \text{for } i_1 < i_2 , \tag{6}$$

where each entry of the upper-triangular tensor is estimated using a sample average and the remaining entries are calculated to ensure the symmetry relations implied by the inclusion $\widehat{\mathbf{P}}_n \in \mathcal{P}_{d,k}$.

As mentioned before, we are interested in computing the solution $\pi^* := \pi(\mathbf{P}, S, \| \cdot \|)$ to the optimization problem (4), but with access only to samples from the passive observation model. For any estimator $\widehat{\pi} \in \Delta_d$ obtained from these samples, we evaluate its error based on its value with respect to the tensor $\mathbf{P}$, i.e.,

$$\Delta_{\mathbf{P}}(\widehat{\pi}, \pi) := v(\widehat{\pi}; S, \mathbf{P}, \| \cdot \|) - v(\pi^*; S, \mathbf{P}, \| \cdot \|). \tag{7}$$

Note that the error $\Delta_{\mathbf{P}}$ implicitly also depends on the set $S$ and the norm $\| \cdot \|$, but we have chosen our notation to be explicit only in the preference tensor $\mathbf{P}$. For the rest of this section, we restrict our attention to convex target sets $S$ and refer them to as *valid sets*. Having established the background, we are now ready to provide sample complexity bounds on the estimation error $\Delta_{\mathbf{P}}(\widehat{\pi}, \pi^*)$.

## 3.1 Upper bounds on the error of the plug-in estimator

While, our focus in this section is to provide upper bounds on the error of the plug-in estimator $\widehat{\pi}_{\text{plug}} = \pi(\widehat{\mathbf{P}}, S, \| \cdot \|)$, we first state a general perturbation bound which relates the error of the optimizer $\pi(\widetilde{\mathbf{P}}, S, \| \cdot \|)$ to the deviation of the tensor $\widetilde{\mathbf{P}}$ from the true tensor $\mathbf{P}$. We use $\mathbf{P}(\cdot, i) \in [0, 1]^{d \times k}$ to denote a matrix formed by viewing the $i$-th slice of $\mathbf{P}$ along its second dimension.

**Theorem 1.** *Suppose the distance $\rho$ is induced by the norm $\| \cdot \|_q$ for some $q \geq 1$. Then for each valid target set $S$ and preference tensor $\widetilde{\mathbf{P}}$, we have*

$$\Delta_{\mathbf{P}}(\pi(\widetilde{\mathbf{P}}), \pi^*) \leq 2 \max_{i \in [d]} \|\widetilde{\mathbf{P}}(\cdot, i) - \mathbf{P}(\cdot, i))\|_{\infty, q}. \tag{8}$$

Note that this theorem is entirely deterministic: it bounds the deviation in the optimal solution to the problem (4) as a function of perturbations to the tensor $\mathbf{P}$. It also applies *uniformly* to all valid target sets $S$. In particular, this result generalizes the perturbation result of Dudik et al. [22, Lemma 3] which obtained such a deviation bound for the single criterion problem with $\pi^*$ as the von Neumann winner. Indeed, one can observe that by setting the distance $\rho(u, v) = |u - v|$ in Theorem 1 for the uni-criterion setup, we have the error $\Delta_{\mathbf{P}}(\pi(\widetilde{\mathbf{P}}), \pi^*) \leq 2\|\widetilde{\mathbf{P}} - \mathbf{P}\|_{\infty, \infty}$, matching the bound of [22].

Let us now illustrate a consequence of this theorem by specializing it to the plug-in estimator, and with the distances given by the $\ell_\infty$ norm.

**Corollary 1.** *Suppose that the distance $\rho$ is induced by the $\ell_\infty$-norm $\| \cdot \|_\infty$. Then there exists a universal constant $c > 0$ such that given a sample size $n > cd^2k \log(\frac{cdk}{\delta})$, we have for each valid target set $S$*

$$\mathbb{E}\left[\Delta_{\mathbf{P}}(\widehat{\pi}_{\text{plug}}, \pi^*)\right] \leq c\sqrt{\frac{d^2k}{n} \log\left(\frac{cdk}{\delta}\right)}, \tag{9}$$

*with probability greater than $1 - \delta$.*

The bound (9) implies that the plug-in estimator $\widehat{\pi}_{\text{plug}}$ is an $\epsilon$-approximate solution whenever the number of samples scales as $n = \widetilde{O}(\frac{d^2k}{\epsilon^2})$. Observe that this sample complexity scales quadratically in the number of objects $d$ and linearly in the number of criteria $k$. This scaling represents the effective dimensionality of the problem instance, since the underlying preference tensor $\mathbf{P}$ has $O(d^2k)$ unknown parameters. Notice that the corollary holds for sample size $n = \widetilde{O}(d^2k)$; this should not be thought of as restrictive, since otherwise, the bound (9) is vacuous.

## 3.2 Information-theoretic lower bounds

While Corollary 1 provides an upper bound on the error of the plug-in estimator that holds *for all* valid target sets $S$, it is natural to ask if this bounds is sharp, i.e., whether there is indeed a target set $S$ for which one can do no better than the plug-in estimator. In this section, we address this question by providing lower bounds on the minimax risk $\mathfrak{M}_{n,d,k}(S, \| \cdot \|_\infty) := \inf_{\widehat{\pi}} \sup_{\mathbf{P} \in \mathcal{P}} \mathbb{E}\left[\Delta_{\mathbf{P}}(\widehat{\pi}, \pi^*)\right]$, where the infimum is taken over *all* estimators that can be computed from $n$ samples from our observation model. It is important to note that the error $\Delta_{\mathbf{P}}$ is computed using the $\ell_\infty$ norm and for the set $S$. Our lower bound will apply to the particular choice of target set $S_0 = [1/2, 1]^k$.

**Theorem 2.** *There is a universal constant $c$ such that for all $d \geq 4$, $k \geq 2$, and $n \geq cd^4 k$, we have*

$$\mathfrak{M}_{n,d,k}(S_0, \|\cdot\|_\infty) \geq c\sqrt{\frac{d^2 k}{n}}. \tag{10}$$

Comparing equations and (9) and (10), we see that for the $\ell_\infty$-norm and the set $S_0$, we have provided upper and lower bounds that match up to a logarithmic factor in the dimension. Thus, the plug-in estimator is indeed optimal for this pair ($\|\cdot\|_\infty$, $S_0$). Further, observe that the above lower bound is non-asymptotic, and holds for all values of $n \gtrsim d^4 k$. This condition on the sample size arises as a consequence of the specific packing set used for establishing the lower bound, and improving it is an interesting open problem.

However, this raises the question of whether the set $S_0$ is special, or alternatively, whether one can obtain an $S$-dependent lower bound. The following proposition shows that at least *asymptotically*, the sample complexity for *any* polyhedral set $S$ obeys a similar lower bound.

**Proposition 2** (Informal). *Suppose that we have a valid polyhedral target set $S$, and that $d \geq 4$. There exists a positive integer $n_0(d, k, S)$ such that for all $n \geq n_0(d, k, S)$ we have*

$$\mathfrak{M}_{n,d,k}(S, \|\cdot\|_\infty) \gtrsim \sqrt{\frac{d^2 k}{n}}. \tag{11}$$

We defer the formal statement and proof of this proposition to Appendix B. This proposition establishes that the plugin estimator $\widehat{\pi}_{\mathsf{plug}}$ is indeed asymptotically optimal in the $\ell_\infty$ norm for broad class of sets $S$.

### 3.3 Computing the plug-in estimator

In the last few sections, we discussed the statistical properties of the plug-in estimator, and showed that its sample complexity was optimal in a minimax sense. We now turn to the algorithmic question: how can the plug-in estimator $\widehat{\pi}_{\mathsf{plug}}$ be computed? Our main result in this direction is the following theorem that characterizes properties of the objective function $v(\pi; \mathbf{P}, S, \|\cdot\|)$.

**Theorem 3.** *Suppose that the distance function is given by an $\ell_q$ norm $\|\cdot\|_q$ for some $q \geq 1$. Then for each valid target set $S$, the objective function $v(\pi; \mathbf{P}, S, \|\cdot\|_q)$ is convex in $\pi$, and Lipschitz in the $\ell_1$ norm, i.e.,*

$$|v(\pi_1; \mathbf{P}, S, \|\cdot\|_q) - v(\pi_2; \mathbf{P}, S, \|\cdot\|_q)| \leq k^{\frac{1}{q}} \cdot \|\pi_1 - \pi_2\|_1 \text{ for each } \pi_1, \pi_2 \in \Delta_d.$$

Theorem 3 establishes that the plug-in estimator can indeed be computed as the solution to a (constrained) convex optimization problem. In Appendix C, we discuss a few specific algorithms based on zeroth-order and first-order methods for obtaining such a solution and an analysis of the corresponding iteration complexity for these methods; see Propositions 5 and 6 in the appendix.

## 4 Autonomous driving user study

In order to evaluate the proposed framework, we applied it to an autonomous driving environment. The objective is to study properties of the randomized policies obtained by our multi-criteria framework—the Blackwell winner for specific choices of the target set—and compare them with the alternative approaches of linear combinations of criteria and the single-criterion (overall) von Neumann winner. We briefly describe the components of the experiment here; see Appendix D for more details.

**Self-driving Environment.** Figure 1(a) shows a snapshot of one of the worlds in this environment with the autonomous car shown in orange. We construct three different worlds in this environment:

W1: The first world comprises an empty stretch of road with no obstacles (20 steps).

W2: The second world consists of a sequence of cones placed in certain sequences (80 steps).

W3: The third world has additional cars driving at varying speeds in their fixed lanes (80 steps).

**Policies.** For our *base policies*, we design five different reward functions encoding different self-driving behaviors. These polices, named Policy A-E, are then set to be the model predictive control based policies based on these reward functions wherein we fix the planning horizon to 6. We defer the details of these reward functions to Appendix D. A *randomized policy* $\pi \in \Delta_5$ is given by a distribution over the base policies A-E. Such a randomized policy is implemented in our environment by randomly sampling a base policy from the mixture distribution after every $H = 18$ time steps and executing this selected policy for that duration. To account for the randomization, we execute each such policy for 5 independent runs in each of the worlds and record these behaviors.

**Subjective Criteria.** We selected five subjective criteria to compare the policies, with questions asking which of the two policies was C1: Less aggressive, C2: More predictable, C3: More quick, C4: More conservative, and had C5: Less collision risk. Such a framing of question ensures that higher score value along any of C1-C5 is preferred; thus a higher score along C1 would imply less aggressive while along C2 would mean more predictable. In addition to the these base criteria, we also consider an *Overall Preference* which compares any pair of policies in an aggregate manner. Additionally, we also asked the users to rate the importance of each criterion in their overall preference.

**Main Hypotheses.** The central focus of the main hypotheses is on comparing the randomized policies given by the Blackwell winner, the overall von Neumann winner, and those given by weighing the criteria linearly.

MH1 There exists a set $S$ such that the Blackwell winner with respect to $S$ and $\ell_\infty$-norm produced by our framework outperforms the overall von Neumann winner.

MH2 The Blackwell winner for oblivious score sets $S$ outperforms both oblivious[2] and data-driven weights for linear combination of criteria.

**Independent Variables.** The independent variable of our experiment is the choice of algorithms for producing the different randomized winners. These comprise the von Neumann winner based on overall comparisons, Blackwell winners based on two oblivious target sets, and 9 different linear combinations weights (3 data-driven and 6 oblivious).

We begin with the two target sets $S_1$ and $S_2$ for our evaluation of the Blackwell winner which were selected in a data-oblivious manner. Set $S_1$ is an axis-aligned set promoting the use of safer policies with score vector constrained to have a larger value along the collision risk axis. Similar to Figure 2(b), the set $S_2$ adds a linear constraint along aggressiveness and collision risk. This target set thus favors policies which are less aggressive and have lower collision risk. For evaluating hypothesis MH2, we considered several weight vectors, both oblivious and data-dependent, comprising average of the users' self-reported weights, that obtained by regressing the overall criterion on C1-C5, and a set of oblivious weights. See Appendix D for details of the sets $S_1$ and $S_2$, and the weights $w_{1:9}$.

**Data collection.** The experiment was conducted in two phases, both of which involved human subjects on Amazon Mechanical Turk (Mturk) (see Appendix D for an illustration of the questionnaire). The first phase of the experiment involved preference elicitation for the five base policies A-E. Each user was asked to provide comparison data for all ten combinations of policies. The cumulative comparison data is given in Appendix D, and the average weight vector elicited from the users was found to be $w_1 = [0.21, 0.19, 0.20, 0.18, 0.22]$. We ran this study with 50 subjects.

In the overall preference elicitation, we saw an approximate ordering amongst the base policies: C $\succ$ E $\succsim$ D $\succsim$ B $\succ$ A. Thus, Policy C was the von Neumann winner along the overall criterion. For each of the linear combination weights $w_1$ through $w_9$, Policy C was the weighted winner. The Blackwell winners R1 and R2 for the sets $S_1$ and $S_2$ with the $\ell_\infty$ distance were found to be R1 $= [0.09, 0.15, 0.30, 0.15, 0.31]$ and R2 $= [0.01, 0.01, 0.31, 0.02, 0.65]$.

In the second phase, we obtained preferences from a set of 41 subjects comparing the randomized polices R1 and R2 with the baseline policies A-E. The results are aggregated in Table 1 in Appendix D.

**Analysis for main hypotheses.** Given that the overall von Neumann winner and those corresponding to weights $w_{1:9}$ were all Policy C, hypotheses MH1 and MH2 reduced whether users prefer at least one of {R1, R2} to the deterministic policy C, that is whether $\mathbf{P}_{ov}(C, R1) < 0.5$ or $\mathbf{P}_{ov}(C, R2) < 0.5$.

Policies C and E were preferred to R1 by 0.71 and 0.61 fraction of the respondents, respectively. On the other hand, R2 was preferred to the von Neumann winner C by 0.66 fraction of the subjects. Using the data, we conducted a hypothesis test with the null and alternative hypotheses given by

$$H_0 : \mathbf{P}_{\mathsf{ov}}(\mathrm{C}, \mathrm{R2}) \geq 0.5, \quad \text{and} \quad H_1 : \mathbf{P}_{\mathsf{ov}}(\mathrm{C}, \mathrm{R2}) < 0.5.$$

Among the hypotheses that make up the (composite) null, our samples have the highest likelihood for the distribution $\mathsf{Ber}(0.5)$. We therefore perform a one-sided hypothesis test with the Binomial distribution with number of samples $n = 41$, success probability $p = 0.5$ and number of successes $x = 14$ (indicating number of subjects which preferred Policy C to R2). The p-value for this test was obtained to be $0.0298$. This supports both our claimed hypotheses MH1 and MH2.

## 5  Discussion and future work

In this paper, we considered the problem of eliciting and learning from preferences along multiple criteria, as a way to obtain rich feedback under weaker assumptions. We introduced the notion of a Blackwell winner, which generalizes many known winning solution concepts. We showed that the Blackwell winner was efficiently computable from samples with a simple and optimal procedure, and also that it outperformed the von Neumann winner in a user study on autonomous driving. Our work raises many interesting follow-up questions: How does the sample complexity vary as a function of the preference tensor $\mathbf{P}$? Can the process of choosing a good target set be automated? What are the analogs of our results in the setting where pairwise comparisons can be elicited *actively*?

## Broader impact

An important step towards deploying AI systems in the real world involves aligning their objectives with human values. Examples of such objectives include safety for autonomous vehicles, fairness for recommender systems, and effectiveness of assistive medical devices. Our paper takes a step towards accomplishing this goal by providing a framework to aggregate human preferences along such subjective criteria, which are often hard to encode mathematically. While our framework is quite expressive and allows for non-linear aggregation across criteria, it leaves the choice of the target set in the hands of the designer. As a possible negative consequence, getting this choice wrong could lead to incorrect inferences and unexpected behavior in the real world.

## Acknowledgments and Disclosure of Funding

We would like to thank Niladri Chatterji, Robert Kleinberg and Karthik Sridharan for helpful discussions, and Andreea Bobu, Micah Carroll, Lawrence Chan and Gokul Swamy for helping with the user study setup.

AP is supported by a Swiss Re research fellowship at the Simons Institute for the Theory of Computing and KB is supported by a JP Morgan AI Fellowship. This work was partially supported by Office of Naval Research Young Investigator Award and a AFOSR grant to ADD, and by Office of Naval Research Grant DOD ONR-N00014-18-1-2640 to MJW.

Additional revenue: ADD is employed as a consultant at Waymo, LLC and PLB is employed as a consultant at Google.

## Footnotes

[1]We let $\Delta_d$ denote the $d$-dimensional simplex.

[2] We use the term oblivious to denote variables that were *fixed* before the data collection phase and data-driven to denote those which are based on collected data.

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
