[Supplementary Material]

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

## A  Blackwell's approachability

Blackwell [11] introduced the concept of approachability as a generalization of the minimax theorem to vector-valued payoffs. Formally, a Blackwell game is an extension of two-player zero-sum games with vector-valued reward functions.

Let $\mathcal{X}, \mathcal{Y}$ denote the action spaces for the two players and $r : \mathcal{X} \times \mathcal{Y} \mapsto \mathbb{R}^k$ be the corresponding vector-valued reward function. Further, let $S \subseteq \mathbb{R}^k$ denote a target set. The objective of player 1 is to ensure that the reward vector $r$ lies in the set $S$ while that of player 2 is ensure that the reward $r$ lies outside this set $S$. Following [1], we introduce the notion of satisfiability and response-satisfiability.

**Definition 1** (Satisfiability). *For a Blackwell game parameterized by* $(\mathcal{X}, \mathcal{Y}, r, S)$*, we say that,*

- *$S$ is* satisfiable *if there exists $x \in \mathcal{X}$ such that for all $y \in \mathcal{Y}$, we have that $r(x, y) \in S$.*

- *$S$ is* response-satisfiable *if for every $y \in \mathcal{Y}$, there exists $x \in \mathcal{X}$ such that $r(x, y) \in S$.*

In the case of scalar rewards, Von Neumann's minimax theorem indicates that any set which is satisfiable is also response-satsifiable. In other words, there exists a strategy for Player 1, oblivious of Player 2's strategy which ensures that the reward belongs to the set $S$ if the set $S$ is response-satisfiable. The existence of such a relation was crucial in obtaining the concept of the Von Neumann winner described in Section 2 for the uni-criterion problem.

However, such a statement fails to hold in the general vector-valued case (see [1] for a counterexample). In order to overcome this limitation, Blackwell [11] defined the notion of approachability as follows.

**Definition 2** (Blackwell's Approachability). *Given a Blackwell game* $(\mathcal{X}, \mathcal{Y}, r, S)$*, we say that a set $S$ is approachable if there exists an algorithm $\mathcal{A}$ which selects points in $\mathcal{X}$ such that for any sequence $y_1, \ldots y_t \in \mathcal{Y}$,*

$$\lim_{T \to \infty} \rho \left( \frac{1}{T} \sum_{t=1}^{T} r(x_t, y_t), S \right) \to 0 \, ,$$

*where $x_t = \mathcal{A}(y_1, \ldots, y_{t-1})$ is the algorithm's play at time $t$ for some distance function $\rho$.*

The celebrated Blackwell's theorem then claims that any set $S$ is approachable iff it is response-satisfiable. This means that while no single choice of action in the set $\mathcal{X}$ can guarantee a response in the set $S$, there exists an algorithm which ensures that in the repeated game, the average rewards approach the set $S$, for any choice of opponent play.

Note that our definition of *achievability* is a stronger requirement than Blackwell's approachability. While approachability requires the time-averaged payoff in a repeated game to belong to the pre-specified set $S$, achievability requires the same to be true in a single-shot play of the game. Indeed, as the following lemma shows, one can construct examples of multi-criteria preference problems which are approachable but not achievable.

**Proposition 3** (Approachability does not imply achievability). *There exists a preference tensor $\mathbf{P} \in \mathcal{P}_{d,k}$ and a target set $S \subset [0,1]^k$ such that*
*a) For the Blackwell game given by $(\Delta_d, \Delta_d, \mathbf{P}, S)$, the set $S$ is approachable, and*
*b) The set $S$ is not achievable with respect to $\mathbf{P}$.*

*Proof.* We will consider an example in a 2-dimensional action space with 2 criteria. Consider the preference matrix given by:

$$\mathbf{P}^1 = \begin{bmatrix} \frac{1}{2} & 1 \\ 0 & \frac{1}{2} \end{bmatrix} \qquad \text{and} \qquad \mathbf{P}^2 = \begin{bmatrix} \frac{1}{2} & 0 \\ 1 & \frac{1}{2} \end{bmatrix} \, , \tag{12}$$

along with the convex set $S = [\frac{1}{2}, 1]^2$. The tensor $\mathbf{P}$ represents the strongest possible trade-off between the two objects: Object 1 is preferred over 2 along the first criterion while the reverse is true for the second criterion.

The Blackwell game given by $(\Delta_d, \Delta_d, \mathbf{P}, S)$ can indeed be shown to be approachable. The set $S$ is response-satisfiable since for every strategy $y \in \Delta_d$ chosen by the column player, the choice of

$x = y$ would yield a reward vector $\mathbf{P}(x, y) = [\frac{1}{2}, \frac{1}{2}] \in S$. Then, by Blackwell's theorem [11], the set $S$ is approachable.

In contrast, consider any choice of distribution $\pi_1 = [p, 1 - p]$ for the multi-criteria preference problem. The corresponding score vectors for responses $i_2 = 1, 2$ are given by:

$$r_1 = \mathbf{P}(\pi_1, i_2 = 1) = \left[\frac{p}{2}, 1 - \frac{p}{2}\right] \qquad \text{and} \qquad r_2 = \mathbf{P}(\pi_1, i_2 = 2) = \left[\frac{1}{2} + \frac{p}{2}, \frac{1}{2} - \frac{p}{2}\right].$$

For any choice of the parameter $p \in [0, 1]$, one cannot have both $r_1$ and $r_2$ simultaneously belong to the set $S$. Hence, we have that the set $S$ is not achievable with respect to $\mathbf{P}$.

This example can be extended to any arbitrary dimension $k$ by extending the tensor to have $\mathbf{P}^j$ equal to the all-half matrix for any criterion $j > 2$ and the target set to be $S = [\frac{1}{2}, 1]^k$. Similarly, in order to extend the example to any dimension, consider the preference tensor (for $k = 2$)

$$\mathbf{P}_d^1 = \begin{bmatrix} \mathbf{P}^1 & \mathbf{P}_{1/2} & \cdots & \mathbf{P}_{1/2} \\ \mathbf{P}_{1/2} & \mathbf{P}^1 & \cdots & \mathbf{P}_{1/2} \\ \vdots & \cdots & \ddots & \vdots \\ \mathbf{P}_{1/2} & \mathbf{P}_{1/2} & \cdots & \mathbf{P}^1 \end{bmatrix} \quad \text{and} \quad \mathbf{P}_d^2 = \begin{bmatrix} \mathbf{P}^2 & \mathbf{P}_{1/2} & \cdots & \mathbf{P}_{1/2} \\ \mathbf{P}_{1/2} & \mathbf{P}^2 & \cdots & \mathbf{P}_{1/2} \\ \vdots & \cdots & \ddots & \vdots \\ \mathbf{P}_{1/2} & \mathbf{P}_{1/2} & \cdots & \mathbf{P}^2 \end{bmatrix},$$

with the smaller matrices $\mathbf{P}^1$ and $\mathbf{P}^2$ from equation (12) at the diagonal and $\mathbf{P}_{1/2}$ denoting the all-half tensor of the appropriate dimension. A similar calculation as for the $d = 2$ case yields that the set $S$ is not achievable. This establishes the required claim. □

## B  Proof of main results

In this section, we provide formal proofs of all the results stated in the main paper. Appendix C to follow collects some additional results and their proofs.

### B.1  Proof of Proposition 1

We establish both parts of the proposition separately.

#### B.1.1  Proof of part (a)

For any weight vector $w \in \Delta_k$, consider the set

$$S_w = \left\{ r \in [0, 1]^k \mid \langle w, r \rangle \geq 1/2 \right\}.$$

The set $S_w$ is clearly convex. Indeed, for any two vectors $r_1, r_2 \in S_w$ and any scalar $\alpha \in [0, 1]$, we have

$$\langle w, \alpha r_1 + (1 - \alpha) r_2 \rangle = \alpha \langle w, r_1 \rangle + (1 - \alpha) \langle w, r_2 \rangle \in \left[\frac{1}{2}, 1\right].$$

It is straightforward to verify that the set $S_w$ is also monotonic with respect to the orthant ordering.

We now show that a von Neumann winner $\pi^*$ of the (single-criterion) preference matrix $\mathbf{P}_w := \mathbf{P}(w)$ can be written as $\pi(\mathbf{P}, S_w, \|\cdot\|)$ for an arbitrary choice of norm $\|\cdot\|$. For each $\widetilde{\pi} \in \Delta_d$, we have

$$\langle w, \mathbf{P}(\pi^*, \widetilde{\pi}) \rangle = \sum_{j \in [k]} w_j \mathbf{P}^j(\pi^*, \widetilde{\pi}) = \mathbf{P}_w(\pi^*, \widetilde{\pi}) \overset{\text{(i)}}{\geq} \frac{1}{2},$$

where the inequality (i) follows since $\pi^*$ is a von Neumann winner for the matrix $\mathbf{P}_w$. Thus, we have the inclusion $\mathbf{P}(\pi^*, \widetilde{\pi}) \in S_w$ for all $\widetilde{\pi} \in \Delta_d$, so that $\max_{\widetilde{\pi} \in \Delta_d} \rho(\mathbf{P}(\pi^*, \widetilde{\pi}), S_w) = 0$ for any distance metric $\rho$. Consequently, we have

$$\pi^* \in \operatorname*{argmin}_{\pi \in \Delta_k} \max_{\widetilde{\pi} \in \Delta_d} \rho(\mathbf{P}(\pi, \widetilde{\pi}), S_w),$$

which establishes the claim for part (a). □

### B.1.2 Proof of part (b)

Consider the multi-criteria preference instance given by target set $S = [\frac{1}{2}, 1]^k$, the $\ell_\infty$ distance function and the preference tensor $\mathbf{P}$

$$\mathbf{P}^1 = \begin{bmatrix} \frac{1}{2} & 1 \\ 0 & \frac{1}{2} \end{bmatrix}, \quad \mathbf{P}^2 = \begin{bmatrix} \frac{1}{2} & 0 \\ 1 & \frac{1}{2} \end{bmatrix}, \quad \text{and} \quad \mathbf{P}^j = \begin{bmatrix} \frac{1}{2} & \frac{1}{2} \\ \frac{1}{2} & \frac{1}{2} \end{bmatrix}$$

The *unique* Blackwell winner for this instance $(\mathbf{P}, S, \|\cdot\|_\infty)$ is given by

$$\underbrace{\pi(\mathbf{P}, S, \|\cdot\|_\infty)}_{\pi^*} = [1/2, 1/2]. \tag{13}$$

For any weight $w \in [0,1]^k$, consider the von Neumann winners corresponding to the weighted matrices $\mathbf{P}_w$

$$\pi(\mathbf{P}_w, [1/2, 1], |\cdot|) = \begin{cases} [1, 0] & \text{for } w \text{ s.t. } \mathbf{P}_w(1, 2) > 0.5 \\ [0, 1] & \text{for } w \text{ s.t. } \mathbf{P}_w(1, 2) < 0.5 \\ \pi \in \Delta_2 & \text{otherwise} \end{cases} \tag{14}$$

Comparing equations (13) and (14) establishes the required claim. $\qquad\square$

### B.2  Proof of Theorem 1

Let us use the shorthand $\widetilde{\pi} := \pi(\widetilde{\mathbf{P}})$. We begin by decomposing the desired error term as

$$\Delta_{\mathbf{P}}(\widetilde{\pi}, \pi^*)$$
$$= \underbrace{v(\widetilde{\pi}; S, \mathbf{P}, \|\cdot\|) - v(\widetilde{\pi}; S, \widetilde{\mathbf{P}}, \|\cdot\|)}_{\text{Perturbation error at } \widetilde{\pi}} + \underbrace{v(\widetilde{\pi}; S, \widetilde{\mathbf{P}}, \|\cdot\|) - v(\pi^*; S, \widetilde{\mathbf{P}}, \|\cdot\|)}_{\leq 0} + \underbrace{v(\pi^*; S, \widetilde{\mathbf{P}}, \|\cdot\|) - v(\pi^*; S, \mathbf{P}, \|\cdot\|)}_{\text{Perturbation error at } \pi^*}$$

In order to obtain a bound on the perturbation errors, note that for any distribution $\pi$, we have

$$v(\pi; S, \mathbf{P}, \|\cdot\|) - v(\pi; S, \widetilde{\mathbf{P}}, \|\cdot\|) = \max_{i_1}[\rho(\mathbf{P}(\pi, i_1), S)] - \max_{i_2}[\rho(\widetilde{\mathbf{P}}(\pi, i_2), S)]$$
$$\overset{(i)}{\leq} \max_{i}[\rho(\mathbf{P}(\pi, i), S) - \rho(\widetilde{\mathbf{P}}(\pi, i), S)], \tag{15}$$

where step (i) follows by setting the $i_2$ equal to $i_1$. Noting that the distance is given by the $\ell_q$ norm, we have

$$v(\pi; S, \mathbf{P}, \|\cdot\|) - v(\pi; S, \widetilde{\mathbf{P}}, \|\cdot\|) \leq \max_{i}[\min_{z_1 \in S} \|\mathbf{P}(\pi, i) - z_1\|_q - \min_{z_2 \in S} \|\widetilde{\mathbf{P}}(\pi, i) - z_2\|_q]$$
$$\overset{(i)}{\leq} \max_{i}[\|\mathbf{P}(\pi, i) - \widetilde{\mathbf{P}}(\pi, i)\|_q],$$

where the inequality (i) follows by setting $z_2$ equal to $z_1$. Taking a supremum over all distributions $\pi$ completes the proof. $\qquad\square$

### B.3  Proof of Corollary 1

By Theorem 1, it suffices to provide a bound on the quantity $\max_i \|\mathbf{P}(\cdot, i) - \widehat{\mathbf{P}}(\cdot, i))\|_{\infty,\infty}$ for the plug-in preference tensor $\widehat{\mathbf{P}}$. Now by definition, we have

$$\max_{i} \|\mathbf{P}(\cdot, i) - \widehat{\mathbf{P}}(\cdot, i))\|_{\infty,\infty} = \max_{i_1, i_2, j} |\mathbf{P}^j(i_1, i_2) - \widehat{\mathbf{P}}^j(i_1, i_2)| .$$

For each $i = (i_1, i_2, j)$ representing some index of the tensor, let $N_i := \#\{\ell \mid \eta_\ell = i\}$ denote the number of samples observed at that index. Since $N_i$ can be written as a sum of i.i.d. Bernoulli random variables, applying the Hoeffding bound yields

$$\Pr\left\{\left|N_i - \frac{n}{d^2 k}\right| \geq c\sqrt{\frac{n \log(c/\delta)}{d^2 k}}\right\} \leq \delta \text{ for each } \delta \in (0, 1).$$

Note that we also have $n \geq c_0 d^2 k \log(c_1 d/\delta)$ by assumption. For a large enough choice of the constants $(c_0, c_1)$, applying the union bound yields the sequence of sandwich relations

$$\frac{n}{2d^2 k} \leq N_i \leq \frac{3n}{2d^2 k} \quad \text{for all indices } i \text{ with probability greater than } 1 - \delta. \tag{16}$$

Furthermore, conditioned on $N_i$ (for $i = (i_1, i_2, j)$), the Hoeffding bound yields the relation

$$\Pr\left\{ |\mathbf{P}^j(i_1, i_2) - \widehat{\mathbf{P}}^j(i_1, i_2)| \geq c\sqrt{\frac{\log(c/\delta)}{N_i}} \right\} \leq \delta \text{ for each } \delta \in (0, 1).$$

Putting this together with a union bound, we have

$$\Pr\left\{ \max_{i_1, i_2, j} |\mathbf{P}^j(i_1, i_2) - \widehat{\mathbf{P}}^j(i_1, i_2)| \geq c\sqrt{\frac{\log(cd^2 k/\delta)}{\min_i N_i}} \right\} \leq \delta. \tag{17}$$

Combining inequalities (16) and (17) with a final union bound completes the proof. $\qquad\square$

### B.4 Proof of Theorem 2

Suppose throughout that $k \geq 2$, and recall the axis-aligned convex target set $S_0 = [\frac{1}{2}, 1]^k$. We split our proof into two cases depending on whether $d$ is even or odd.

**Case 1: $d$ even.** We use Le Cam's method and construct two problem instances with preference tensors given by $\mathbf{P}_0$ and $\mathbf{P}_1$. Two key elements in the construction are the following $2 \times 2 \times 2$ tensors, which we denote by $\mathbf{P}_{\mathsf{cr}}$ and $\widetilde{\mathbf{P}}_{\mathsf{cr}}$, respectively. Their entries are given by

$$\mathbf{P}_{\mathsf{cr}}^1 = \begin{bmatrix} \frac{1}{2} & \frac{1}{2} + \gamma \\ \frac{1}{2} - \gamma & \frac{1}{2} \end{bmatrix} \quad , \quad \mathbf{P}_{\mathsf{cr}}^2 = \begin{bmatrix} \frac{1}{2} & \frac{1}{2} - \gamma \\ \frac{1}{2} + \gamma & \frac{1}{2} \end{bmatrix},$$

$$\widetilde{\mathbf{P}}_{\mathsf{cr}}^1 = \begin{bmatrix} \frac{1}{2} & \frac{1}{2} + \frac{\gamma}{d} \\ \frac{1}{2} - \frac{\gamma}{d} & \frac{1}{2} \end{bmatrix} \quad \text{and} \quad \widetilde{\mathbf{P}}_{\mathsf{cr}}^2 = \begin{bmatrix} \frac{1}{2} & \frac{1}{2} - \frac{\gamma}{d} \\ \frac{1}{2} + \frac{\gamma}{d} & \frac{1}{2} \end{bmatrix}.$$

Note that these tensors are parameterized by a scalar $\gamma \in [0, 1/2]$, whose exact value we specify shortly. Also denote by $\mathbf{P}_{1/2}$ the $2 \times 2 \times 2$ all-half tensor. We are now ready to construct the pair of $d \times d \times k$ preference tensors $(\mathbf{P}_0, \mathbf{P}_1)$.

In order to construct tensor $\mathbf{P}_0$, we specify its entries on the first two criteria according to

$$\mathbf{P}_0^{1:2} = \begin{bmatrix} \mathbf{P}_{1/2} & \mathbf{P}_{1/2} & \cdots & \mathbf{P}_{1/2} \\ \mathbf{P}_{1/2} & \mathbf{P}_{\mathsf{cr}} & \cdots & \mathbf{P}_{1/2} \\ \vdots & \cdots & \ddots & \vdots \\ \mathbf{P}_{1/2} & \mathbf{P}_{1/2} & \cdots & \mathbf{P}_{\mathsf{cr}} \end{bmatrix}, \tag{18}$$

and set the entries on the remaining $k - 2$ criteria to $1/2$.

On the other hand, the first two criteria of the tensor $\mathbf{P}_1$ are given by

$$\mathbf{P}_1^{1:2} = \begin{bmatrix} \widetilde{\mathbf{P}}_{\mathsf{cr}} & \mathbf{P}_{1/2} & \cdots & \mathbf{P}_{1/2} \\ \mathbf{P}_{1/2} & \mathbf{P}_{\mathsf{cr}} & \cdots & \mathbf{P}_{1/2} \\ \vdots & \cdots & \ddots & \vdots \\ \mathbf{P}_{1/2} & \mathbf{P}_{1/2} & \cdots & \mathbf{P}_{\mathsf{cr}} \end{bmatrix}, \tag{19}$$

with the entries on the remaining $k - 2$ criteria once again set identically to $1/2$.

Note that the tensors $\mathbf{P}_0$ and $\mathbf{P}_1$ only differ on the first $2 \times 2 \times 2$ block. Furthermore, the following lemma provides an exact calculation of the values $\min_\pi v(\pi; \mathbf{P}_0, S_0, \|\cdot\|_\infty)$ and $\min_\pi v(\pi; \mathbf{P}_1, S_0, \|\cdot\|_\infty)$.

**Lemma 1.** *We have*

$$\mathcal{V}_0 := \min_{\pi} v(\pi; \mathbf{P}_0, S_0, \|\cdot\|_\infty) = 0 \quad and \quad \mathcal{V}_1 := \min_{\pi} v(\pi; \mathbf{P}_0, S_0, \|\cdot\|_\infty) = \frac{\gamma}{3d-2}.$$

Given samples from these two instances, we now use Le Cam's lemma [see 52, Chap 2] to lower bound the minimax risk as

$$\mathfrak{M}_{n,d,k}(S_0, \|\cdot\|_\infty) \geq \frac{|\mathcal{V}_0 - \mathcal{V}_1|}{2}\left(1 - \|\mathbb{P}_0^n - \mathbb{P}_1^n\|_{\mathrm{TV}}\right) = \frac{\gamma}{2(3d-2)}\left(1 - \|\mathbb{P}_0^n - \mathbb{P}_1^n\|_{\mathrm{TV}}\right), \quad (20)$$

where $\mathbb{P}_0^n$ and $\mathbb{P}_1^n$ are the probability distributions induced on sample space by the passive sampling strategy applied to the tensor $\mathbf{P}_0$ and $\mathbf{P}_1$, respectively.

Using Pinsker's inequality, the decoupling property for KL divergence and the fact that that $\mathrm{KL}(P\|Q) \leq \chi^2(P\|Q)$, we have

$$\|\mathbb{P}_0^n - \mathbb{P}_1^n\|_{\mathrm{TV}} \leq \sqrt{\frac{n}{2}\mathrm{KL}(\mathbb{P}_1\|\mathbb{P}_0)} \leq \sqrt{\frac{n}{2}\chi^2(\mathbb{P}_1\|\mathbb{P}_0)}. \quad (21)$$

The chi-squared distance between the two distributions $\mathbb{P}_0$ and $\mathbb{P}_1$ is given by

$$\chi^2(\mathbb{P}_1\|\mathbb{P}_0) = \frac{1}{d^2 k}\sum_{(i_1,i_2,j)}\left(\frac{\mathbf{P}_1^j(i_1,i_2)}{\mathbf{P}_2^j(i_1,i_2)} - 1\right)^2 \overset{(i)}{=} \frac{2}{d^2 k}\left(\left(\frac{2\gamma}{d}\right)^2 + \left(-\frac{2\gamma}{d}\right)^2\right) = \frac{16\gamma^2}{d^4 k},$$

where step (i) follows from the fact that $\mathbf{P}_1$ and $\mathbf{P}_2$ differ only in 4 entries and that the passive sampling strategy samples each index uniformly at random. Putting together the pieces, we have:

$$\mathfrak{M}_{n,d,k}(S_0, \|\cdot\|_\infty) \geq \frac{\gamma}{2(3d-2)}\left(1 - \sqrt{\frac{n}{2}\frac{16\gamma^2}{d^4 k}}\right) \overset{(ii)}{=} \frac{1}{48\sqrt{2}}\sqrt{\frac{d^2 k}{n}}.$$

where step (ii) follows by setting $\gamma^2 = \frac{d^4 k}{32n}$ and using the fact that $3d - 2 \leq 3d$. Note that since we require $\gamma^2 \leq \frac{1}{4}$, the above bound is valid only for $n \gtrsim d^4 k$. This concludes the proof for even $d$.

**Case 2: $d$ odd.** By assumption, we have $d \geq 5$. In this case, we construct $\mathbf{P}_0$ and $\mathbf{P}_1$ exactly as before, but replace $\mathbf{P}_{\mathrm{cr}}$ in the last two rows of both $\mathbf{P}_0$ and $\mathbf{P}_1$ with the following modified $3 \times 3 \times 2$ tensor:

$$\mathbf{P}_{\mathrm{cr},3}^1 = \begin{bmatrix} \frac{1}{2} & \frac{1}{2}+\gamma & \frac{1}{2}-\gamma \\ \frac{1}{2}-\gamma & \frac{1}{2} & \frac{1}{2}-\gamma \\ \frac{1}{2}+\gamma & \frac{1}{2}+\gamma & \frac{1}{2} \end{bmatrix} \quad and \quad \mathbf{P}_{\mathrm{cr},3}^2 = \begin{bmatrix} \frac{1}{2} & \frac{1}{2}-\gamma & \frac{1}{2}+\gamma \\ \frac{1}{2}+\gamma & \frac{1}{2} & \frac{1}{2}+\gamma \\ \frac{1}{2}-\gamma & \frac{1}{2}-\gamma & \frac{1}{2} \end{bmatrix}.$$

By mimicking its proof, it can be verified that this modification ensures that the corresponding values $\mathcal{V}_0$ and $\mathcal{V}_1$ still satisfy Lemma 1. Thus, the lower bound remains unchanged up to constant factors. $\qquad\square$

### B.4.1 Proof of Lemma 1

Let us compute the two values separately.

**Computing $\mathcal{V}_0$.** The choice of distribution $\pi^* = [1, 0, \ldots, 0]$ yields the score vector $[1/2, 1/2, \ldots, 1/2]$, which is in the set $S_0$. Thus, we have $\mathcal{V}_0 = 0$.

**Computing $\mathcal{V}_1$.** Note that the optimal distribution $\pi^*$ achieving the value $\mathcal{V}_1$ will be of the form

$$\pi^* = [p/2, p/2, (1-p)/(d-2), \ldots, (1-p)/(d-2)] \text{ for some } p \in [0,1].$$

This follows from the symmetry in the preference tensor for row objects ranging from 3 to $d$. Given such a distribution $\pi^*$, the distance of the reward vector from the set $S_0$ is given by

$$\inf_{z \in S}\|\mathbf{P}(\pi^*, i_2) - z\|_\infty = \begin{cases} \frac{\gamma p}{2d} & i_2 = 1, 2 \\ \frac{\gamma(1-p)}{d-2} & \text{o.w.} \end{cases}.$$

Thus, for any value of $p > 2d/(3d-2)$, the distance is maximized for $i_2 \in \{1, 2\}$, and yields a value $\gamma p/(2d)$. On the other hand, for $p < 2d/(3d-2)$, the maximizing index is $i_2 \geq 3$, and the maximizing value is $\gamma(1-p)/(d-2)$. Optimizing these values for $p$ yields the claim. $\qquad\square$

## B.5 Instance dependent lower bounds

In this section, we give a formal statement of Proposition 2 along with its proof.

We begin by defining some notation. For any $\alpha, \beta \in [-\frac{1}{2}, \frac{1}{2}]$ and choice of criteria $j_1, j_2 \in [k]$, we define the tensor $\mathbf{P}_{\alpha,\beta}^{(j_1,j_2)} \in [0,1]^{2\times 2\times k}$ as

$$\mathbf{P}_{\alpha,\beta}^{j_1} = \begin{bmatrix} \frac{1}{2} & \frac{1}{2}+\alpha \\ \frac{1}{2}-\alpha & \frac{1}{2} \end{bmatrix}, \quad \mathbf{P}_{\alpha,\beta}^{j_2} = \begin{bmatrix} \frac{1}{2} & \frac{1}{2}+\beta \\ \frac{1}{2}-\beta & \frac{1}{2} \end{bmatrix} \quad \text{and} \quad \mathbf{P}_{\alpha,\beta}^{j} = \begin{bmatrix} \frac{1}{2} & \frac{1}{2} \\ \frac{1}{2} & \frac{1}{2} \end{bmatrix} \text{ for } j \neq \{j_1, j_2\} .$$

Further, we denote by $\mathbf{P}_{1/2}$ the all-half tensor whose dimensions may vary depending on the context. Any distribution $\pi$ over the two objects can be parameterized by a value $q \in [0,1]$ with $q$ being the probability placed on the first object and $1-q$ the probability on the second object. We will consider the distance function given by the $\ell_\infty$ norm. Given this distance function, we overload our notation for the value

$$v(q; \mathbf{P}_{\alpha,\beta}^{(j_1,j_2)}, S) = \max_i[\rho(\mathbf{P}_{\alpha,\beta}^{(j_1,j_2)}(q,i), S)] \quad \text{and} \quad \mathcal{V}(\mathbf{P}_{\alpha,\beta}^{(j_1,j_2)}; S) = \min_q v(q; \mathbf{P}_{\alpha,\beta}^{(j_1,j_2)}; S) .$$
(22)

We now state our main assumption for the score set $S$ which allows us to formulate our lower bound.

**Assumption 1.** *There exists a pair of criteria $(j_1, j_2)$, values $\alpha_0 \in (0, \frac{1}{2}]$ and $\beta_0 \in [-\frac{1}{2}, 0]$, and a gap parameter $\gamma > 0$ such that*

$$\mathcal{V}(\mathbf{P}_{1/2}; S) + \gamma \leq \mathcal{V}(\mathbf{P}_{\alpha_0,\beta_0}^{(j_1,j_2)}; S)$$

*for the all-half tensor $\mathbf{P}_{1/2} \in [0,1]^{2\times 2\times k}$.*

The assumption above indicates that there exists a pair of criteria along which one can observe some sort of trade-off when they interact with the underlying score set $S$. The preference tensor $\mathbf{P}_{\alpha_0,\beta_0}^{(j_1,j_2)}$ captures this trade-off and the gap parameter $\gamma$ quantifies it. Going forward, we assume without loss of generality that $(j_1, j_2) = (1, 2)$ and drop the dependence of the tensor on these indices, writing $\mathbf{P}_{\alpha_0,\beta_0} \equiv \mathbf{P}_{\alpha_0,\beta_0}^{(1,2)}$. The following lemma indicates the importance of the special values of $(\alpha, \beta) = (0, 0)$ for which $\mathbf{P}_{0,0} = \mathbf{P}_{1/2}$.

**Lemma 2.** *For any $\alpha, \beta \in [-\frac{1}{2}, \frac{1}{2}]$, we have $\mathcal{V}(\mathbf{P}_{0,0}; S) \leq \mathcal{V}(\mathbf{P}_{\alpha,\beta}; S)$.*

The above lemma establishes that for any set, the value attained by setting $(\alpha_0, \beta) = (0, 0)$ will be lower than any other setting of the same parameters. For any parameter $\delta \in [0, 1]$, denote by $\mathbf{P}_{\mathsf{wt},\delta}$ the weighted tensor

$$\mathbf{P}_{\mathsf{wt},\delta} := (1-\delta)\mathbf{P}_{0,0} + \delta\mathbf{P}_{\alpha_0,\beta_0}.$$

In order to understand the value $\mathcal{V}(\mathbf{P}_{\mathsf{wt},\delta}; S)$, we establish the following structural lemma which gives us insight into how this value varies as a function of the parameter $\delta \in [0, 1]$.

**Lemma 3.** *Consider a target set $S$ that is given by an intersection of $h$ half-spaces. Then, the value function $\mathcal{V}(\mathbf{P}_{\mathsf{wt},\delta}; S)$ is a piece-wise linear and continuous function of $\delta \in [0, 1]$ with at most $4h$ pieces.*

The above lemma states that the value $\mathcal{V}(\mathbf{P}_{\mathsf{wt},\delta}; S)$ is a piece-wise linear function of $\delta$. Consider the first such piece which has a non-zero slope. Such a line has to exist since $\mathcal{V}(\mathbf{P}_{\mathsf{wt},\delta})$ is continuous in $\delta$ and we have $\mathcal{V}(\mathbf{P}_{\mathsf{wt},0}) < \mathcal{V}(\mathbf{P}_{\mathsf{wt},1})$. Also, this slope has to be positive since we know from Lemma 2 that $\mathcal{V}(\mathbf{P}_{\mathsf{wt},0}) \leq \mathcal{V}(\mathbf{P}_{\mathsf{wt},\delta})$ for any $\delta \in [0, 1]$. Denote the starting point of this line by $\delta_0$ and the corresponding slope by $m_0$, and observe that the value $\mathcal{V}(\mathbf{P}_{\mathsf{wt},\delta_0}) = \mathcal{V}(\mathbf{P}_{\mathsf{wt},0})$. With this notation, we now proceed to prove the lower bound on sample complexity for any polyhedral target score set $S$.

**Proposition 4** (Formal)**.** *Suppose that we have a valid polyhedral target set $S$ satisfying Assumption 1 with parameters $(\alpha_0, \beta_0)$. Then, there exists a universal constant $c$ such that for all $d \geq 4$, $k \geq 2$, and $n \geq \frac{d^2 k}{\delta^2} \frac{(1/2 - \delta_0\alpha_0)^2}{\alpha_0^2 + \beta_0^2}$, we have*

$$\mathfrak{M}_{n,d,k}(S, \|\cdot\|_\infty) \geq c \frac{m_0(\frac{1}{2} - \delta_0\alpha_0)}{\sqrt{\alpha_0^2 + \beta_0^2}} \sqrt{\frac{d^2 k}{n}} .$$
(23)

*Proof.* For this proof, we focus on the case when the number of criteria $k$ is even. The proof for the case when $k$ is odd can be obtained similar to the proof of Theorem 2.

We use Le Cam's method for obtaining a lower bound on the minimax value and construct the lower bound instances using the tensor given by $\mathbf{P}_{\mathsf{wt},\delta}$. For some $\delta \in [0,1]$ (to be fixed later), consider the parameter $\delta_1 = \delta_0 + \delta$. Using these values of $\delta_0$ and $\delta_1$, we create the following two instances $\mathbf{P}_0$ and $\mathbf{P}_1$:

$$
\mathbf{P}_0 = \begin{bmatrix} \mathbf{P}_{\mathsf{wt},\delta_0} & \mathbf{P}_{1/2} & \cdots & \mathbf{P}_{1/2} \\ \mathbf{P}_{1/2} & \mathbf{P}_{\alpha_0,\beta_0} & \cdots & \mathbf{P}_{1/2} \\ \vdots & \cdots & \ddots & \vdots \\ \mathbf{P}_{1/2} & \mathbf{P}_{1/2} & \cdots & \mathbf{P}_{\alpha_0,\beta_0} \end{bmatrix} \quad \text{and} \quad \mathbf{P}_1 = \begin{bmatrix} \mathbf{P}_{\mathsf{wt},\delta_1} & \mathbf{P}_{1/2} & \cdots & \mathbf{P}_{1/2} \\ \mathbf{P}_{1/2} & \mathbf{P}_{\alpha_0,\beta_0} & \cdots & \mathbf{P}_{1/2} \\ \vdots & \cdots & \ddots & \vdots \\ \mathbf{P}_{1/2} & \mathbf{P}_{1/2} & \cdots & \mathbf{P}_{\alpha_0,\beta_0} \end{bmatrix},
$$

where $\mathbf{P}_{\alpha_0,\beta_0}$ is as given by Assumption 1. The following lemma now shows that that there exists a small enough $\bar{\delta}$ such that the value function $\mathcal{V}(\mathbf{P}_{\mathsf{wt},\delta};S)$ is linear in the range $\delta \in [\delta_0,\delta_1]$.

**Lemma 4.** *There exists a $\bar{\delta} \in (0,1)$ such that for all $\delta \in [0,\bar{\delta}]$ and $\delta_1 = \delta_0 + \delta$, we have*

a. *The value $\mathcal{V}(\mathbf{P}_{wt,\delta_1};S) = \mathcal{V}(\mathbf{P}_{wt,\delta_0};S) + \delta m_0$.*

b. *The minimizer $\pi_1^*$ for $\mathbf{P}_1^*$ is given by $\pi_1^* = [q_0, 1 - q_0, 0 \ldots, 0]$.*

We defer the proof of this lemma to the end of the section. Thus, for a small enough value of $\delta \in [0,\bar{\delta}]$, we have $|\mathcal{V}(\mathbf{P}_0) - \mathcal{V}(\mathbf{P}_1)| = \delta m_0$. As was shown in the proof of Theorem 2, the minimax rate is lower bounded as

$$
\mathfrak{M}_{n,d,k}(S, \|\cdot\|_\infty) \geq \frac{|\mathcal{V}(\mathbf{P}_0) - \mathcal{V}(\mathbf{P}_1)|}{2}(1 - \|\mathbb{P}_0^n - \mathbb{P}_1^n\|_{\mathrm{TV}}) \geq \frac{\delta m_0}{2}\left(1 - \sqrt{\frac{n}{2}\chi^2(\mathbb{P}_1\|\mathbb{P}_0)}\right),
$$
(24)

where $\mathbb{P}_0^n$ and $\mathbb{P}_1^n$ are the probability distributions induced on sample space by the passive sampling strategy and the preference tensor $\mathbf{P}_0$ and $\mathbf{P}_1$ respectively. In order to obtain the requisite lower bound, we proceed to compute an upper bound on the chi-squared distance between the two distributions $\mathbb{P}_0$ and $\mathbb{P}_1$ as

$$
\begin{aligned}
\chi^2(\mathbb{P}_1\|\mathbb{P}_0) &= \frac{1}{d^2 k} \sum_{(i_1,i_2,j)} \left(\frac{\mathbf{P}_1^j(i_1,i_2)}{\mathbf{P}_0^j(i_1,i_2)} - 1\right)^2 \\
&\overset{(i)}{\leq} \frac{2}{d^2 k}\left(\left(\frac{\alpha_0^2\delta^2}{(\frac{1}{2} - \delta_0\alpha_0)^2}\right) + \left(\frac{\beta_0^2\delta^2}{(\frac{1}{2} + \delta_0\beta_0)^2}\right)\right) \\
&\overset{(ii)}{\leq} \frac{2\delta^2}{d^2 k}\left(\frac{\alpha_0^2 + \beta_0^2}{(\frac{1}{2} - \delta_0\alpha_0)^2}\right),
\end{aligned}
$$

where (i) follows from the fact that the instances $\mathbf{P}_0$ and $\mathbf{P}_1$ differ only in 4 entries and (ii) follows from the assumption that $|\alpha_0| \geq |\beta_0|$. Now, substituting the value of $\delta^2 = \frac{d^2 k}{4n} \cdot \frac{(\frac{1}{2} - \delta_0\alpha_0)^2}{\alpha_0^2 + \beta_0^2}$ and using the above bound with equation (24), we have

$$
\mathfrak{M}_{n,d,k}(S, \|\cdot\|_\infty) \geq \frac{m_0(\frac{1}{2} - \delta_0\alpha_0)}{8\sqrt{\alpha_0^2 + \beta_0^2}}\sqrt{\frac{d^2 k}{n}},
$$

which holds whenever we have $\delta \in [0,\bar{\delta}]$ or equivalently $n \geq \frac{d^2 k}{4\bar{\delta}^2}\frac{(\frac{1}{2} - \delta_0\alpha_0)^2}{\alpha_0^2 + \beta_0^2}$. This establishes the desired claim. $\qquad\square$

### B.5.1 Proof of Lemma 2

For any $\alpha, \beta \in [-\frac{1}{2}, \frac{1}{2}]$, consider the value

$$
\begin{aligned}
\mathcal{V}(\mathbf{P}_{\alpha,\beta}; S) &= \min_{q \in [0,1]} \max_i [\rho(\mathbf{P}_{\alpha,\beta}(q, i), S)] \\
&= \min_{q \in [0,1]} \max_{\tau \in [0,1]} [\rho(\mathbf{P}_{\alpha,\beta}(q, \tau), S)] \\
&\overset{(i)}{\geq} \rho\left(\left[\frac{1}{2}\right]^k, S\right) = \mathcal{V}(\mathbf{P}_{1/2}; S) ,
\end{aligned}
$$

where (i) follows by setting $\tau = q$ and $\left[\frac{1}{2}\right]^k$ denotes the vector with each entry set to half. This establishes the claim. $\square$

### B.5.2 Proof of Lemma 3

Let us denote by $q_0$ any minimizer of the value $v(q; \mathbf{P}_{\alpha_0,\beta_0}, S)$ and the two score vectors corresponding to the choices for $i$ in equation (22) by $z_{1,i} := \mathbf{P}_{\alpha_0,\beta_0}(q_0, i)$. Observe that for $\mathbf{P}_{\text{wt},\delta}$, the distribution given by $q_0$ is still a minimizer of its value. Further, the score vectors for the two column choices are given by:

$$
z_{\delta,i} = (1 - \delta)\left[\frac{1}{2}\right]^k + \delta z_{1,i} \quad \text{for } i = \{1, 2\}.
$$

Recall that the distance function is given by $\rho(z_{\delta,i}, S) = \min_{z \in S} \|z_{\delta,i} - z\|_\infty$. Now, the minimizer $z$ will lie on the closest hyperplane(s) to the point $z_{\delta,i}$. In order to establish the claim, it suffices to show that for any fixed hyperplane[3] $H$, the distance function given by $\rho(z_{\delta,i}, H)$ is a piece-wise linear function for $\delta \in [0, 1]$.

Let us consider a point $z_{\delta,i}$ which does not belong to the half-space given by $H$, since otherwise, the distance to the halfspace is 0. If we have $\rho(z_{\delta,i}, H) = \zeta$, then the vector $z_{\delta,i} + \zeta 1_k$ must lie on the hyperplane $H$. This follows from the monotonicity property of the hyperplane $H$.

For any $\delta = \frac{1}{2}\delta_1 + \frac{1}{2}\delta_2$ such that $z_{\delta_1,i}$ and $z_{\delta_2,i}$ do not belong to the half-space given by $H$, we have

$$
\rho(z_{\delta,i}) = \frac{1}{2}\underbrace{\rho(z_{\delta_1,i})}_{\zeta_1} + \frac{1}{2}\underbrace{\rho(z_{\delta_2,i})}_{\zeta_2} ,
$$

where the above equality follows since $z_{\delta_1,i} + \zeta_1 1_k$ and $z_{\delta_2,i} + \zeta_2 1_k$ both lie on the hyperplane $H$ and therefore $z_{\delta,i} + \frac{\zeta_1 + \zeta_2}{2} 1_k$ also lies on the hyperplane. Combined with the fact that for any point $z_{\delta,i}$ which lies in the half-space given by $H$, the distance $\rho(z_{\delta,i}, H) = 0$, we have that the function $\rho(z_{\delta,i}, H)$ is a piece-wise linear function with at most 2 linear pieces for $\delta \in [0, 1]$.

Since $\rho(z_{\delta,i}, S)$ is a minimum over $h$ hyperplanes, this function is itself a piece-wise linear function with at most $2h$ pieces. The desired claim now follows from noting that the value function $\mathcal{V}(\mathbf{P}_{\text{wt},\delta}; S)$ is a maximum over two piece-wise linear functions each with at most $2h$ pieces. $\square$

### B.5.3 Proof of Lemma 4

Consider $\delta_1 = \delta_0 + \delta$ such that $\delta_0$ and $\delta_1$ share the same linear piece. This can be guaranteed to hold true for all $\delta \leq \bar{\delta}_1$ by the piecewise linear nature of the value $\mathcal{V}(\mathbf{P}_{\text{wt},\delta})$.

For part (b) of the claim, let us consider the tensor $\tilde{\mathbf{P}} = \mathbf{P}_1(3:, 3:)$ formed by removing the first two rows and columns from the tensor $\mathbf{P}_1$. Then, from Assumption 1, we have that $\mathcal{V}(\tilde{\mathbf{P}}; S) \geq \mathcal{V}(\mathbf{P}_{1/2}; S) + \tilde{\gamma}$ for some $\tilde{\gamma} > 0$. Selecting a value of $\bar{\delta}_2$ such that $\bar{\delta}_2 m_0 \leq \tilde{\gamma}$, we can ensure that condition (b.) is satisfied.

Finally, setting $\bar{\delta} = \min(\bar{\delta}_1, \bar{\delta}_2)$ completes the proof. $\square$

### B.6 Proof of Theorem 3

Let us prove the two claims of the theorem separately. We use the shorthand $v(\pi) := v(\pi; \mathbf{P}, S, \|\cdot\|)$ for convenience.

**Establishing convexity.** Consider any two distributions $\pi_1, \pi_2 \in \Delta_k$ and a scalar $\alpha \in [0, 1]$. Since the set $S$ is closed and convex, we have

$$
\begin{aligned}
v(\alpha\pi_1 + (1-\alpha)\pi_2) &= \max_{i \in [d]} \min_{z \in S} \left[ \rho(\mathbf{P}(\alpha\pi_1 + (1-\alpha)\pi_2, i), z) \right] \\
&\overset{(i)}{=} \max_{i \in [d]} \min_{z_1, z_2 \in S} \left[ \rho(\alpha\mathbf{P}(\pi_1, i) + (1-\alpha)\mathbf{P}(\pi_2, i), \alpha z_1 + (1-\alpha)z_2) \right] \\
&\overset{(ii)}{\leq} \max_{i \in [d]} \left( \alpha \cdot \min_{z_1 \in S} \left[ \rho(\mathbf{P}(\pi_1, i), z_1) \right] + (1-\alpha) \cdot \min_{z_2 \in S} \left[ \rho(\mathbf{P}(\pi_2, i), z_2) \right] \right) \\
&\overset{(iii)}{\leq} \alpha v(\pi_1) + (1-\alpha)v(\pi_2) \,,
\end{aligned}
$$

where (i) follows from the convexity of $S$ and linearity of the preference evaluation (Eq. (2)), (ii) follows from the convexity of the distance function given by $\ell_q$ norm and (iii) follows from distributing the max over the two terms. This establishes the first part of the theorem.

**Establishing the Lipschitz bound.** Consider any two distributions $\pi_1, \pi_2 \in \Delta_d$. The difference in their value function can then be upper bounded as

$$
\begin{aligned}
|v(\pi_1) - v(\pi_2)| &= | \max_{i_1 \in [d]} \left[ \rho(\mathbf{P}(\pi_1, i_1), S) \right] - \max_{i_2 \in [d]} \left[ \rho(\mathbf{P}(\pi_2, i_2), S) \right] | \\
&\overset{(i)}{\leq} \max_{i \in [d]} |\rho(\mathbf{P}(\pi_1, i), S) - \rho(\mathbf{P}(\pi_2, i), S)| \\
&= \max_{i \in [d]} | \min_{z_1 \in S} \rho(\mathbf{P}(\pi_1, i), z_1) - \min_{z_2 \in S} \rho(\mathbf{P}(\pi_2, i), z_2)| \\
&\overset{(ii)}{\leq} \max_{i \in [d]} \max_{z \in S} |\rho(\mathbf{P}(\pi_1, i), z) - \rho(\mathbf{P}(\pi_2, i), z)| \,,
\end{aligned}
$$

where (i) follows from using the inequality $| \max_x f(x) - \max_y g(y)| \leq \max_x |f(x) - g(x)|$ and (ii) follows through a similar inequality $| \min_x f(x) - \min_y g(y)| \leq \max_x |f(x) - g(x)|$. Since the distance function $\rho$ is specified by the $\ell_q$ norm $\|\cdot\|_q$, we have

$$
\begin{aligned}
|v(\pi_1) - v(\pi_2)| &\leq \max_{i \in [d]} \|\mathbf{P}(\pi_1, i) - \mathbf{P}(\pi_2, i)\|_q \\
&= \left[ \sum_{j=1}^{k} \left( \langle \pi_1 - \pi_2, \mathbf{P}^j(\cdot, i) \rangle \right)^q \right]^{\frac{1}{q}} \\
&\overset{(i)}{\leq} k^{\frac{1}{q}} \cdot \|\pi_1 - \pi_2\|_1 \,,
\end{aligned}
$$

where (i) follows from an application of Hölder's inequality ($\ell_1 - \ell_\infty$) to the inner product $\langle \pi_1 - \pi_2, \mathbf{P}^j(\cdot, i) \rangle$ and the fact that $\mathbf{P}^j(i_1, i_2) \in [0, 1]$ for any $(i_1, i_2, j)$. This establishes the Lipschitz bound and concludes the proof of the theorem. $\qquad\square$

## C  Additional results and their proofs

This section covers additional sample complexity results as well as optimization algorithms for finding the Blackwell winner of a multi-criteria preference learning instance.

### C.1  Sample complexity bounds for $\ell_1$ norm

**Corollary 2.** *Suppose that the distance $\rho$ is induced by the $\ell_1$ norm $\|\cdot\|_1$. Then there exists a universal constant $c > 0$ such that given a sample size $n > cd^2 k \log(\frac{cdk}{\delta})$, we have for each valid target set $S$*

$$
\Delta_{\mathbf{P}}(\widehat{\pi}_{\mathsf{plug}}, \pi^*) \leq ck\sqrt{\frac{d^2 k}{n} \log\left(\frac{cdk}{\delta}\right)} \tag{25}
$$

*with probability exceeding $1 - \delta$.*

---
**Algorithm 1:** Zeroth-order method for multi-criteria preference learning
---
**Input:** Time steps $T$, step size $\eta$, smoothing radius $\delta$
**Initialize:** $\theta_1 = 0$
**for** $t = 1, \ldots, T$ **do**
$\quad$ $\pi_t = \operatorname{argmax}_{\pi \in \Delta_d} \langle \theta_t, \pi \rangle - r(\pi)$ where $r(\pi) = \sum_i \pi_i \log(\pi_i)$
$\quad$ Sample $u_t$ uniformly from the Euclidean unit sphere $\{u \mid \|u\|_2 = 1\}$
$\quad$ For every $i \in [d]$, query points $z_{1,i} = \mathbf{P}(\pi_t + \delta u_t, i)$ and $z_{2,i} = \mathbf{P}(\pi_t + \delta u_t, i)$
$\quad$ Set $v(\pi_t + \delta u_t; \mathbf{P}, S, \rho) = \max_i \rho(z_{1,i}, S)$ and $v(\pi_t - \delta u_t; \mathbf{P}, S, \rho) = \max_i \rho(z_{2,i}, S)$
$\quad$ Set sub-gradient estimate $\hat{g}_t = \frac{d}{2\delta} \left( v(\pi_t + \delta u_t; \mathbf{P}, S, \rho) - v(\pi_t - \delta u_t; \mathbf{P}, S, \rho) \right) u_t$
$\quad$ Update $\theta_{t+1} = \theta_t - \eta \hat{g}_t$
**Output:** $\bar{\pi}_T = \frac{1}{T} \sum_{t=1}^{T} \pi_t$
---

*Proof.* Being somewhat more explicit with our notation, let $N_{(i_1, i_2, j)}$ denote the number of samples observed under the passive sampling model at index $(i_1, i_2, j)$ of the tensor. Proceeding as in equation (17), we have

$$\Pr \left\{ \|\mathbf{P}^j(\cdot, i_2) - \widehat{\mathbf{P}}^j(\cdot, i_2)\|_\infty \ge c \sqrt{\frac{\log(cd/\delta)}{\min_{i_1 \in [d]} N_{(i_1, i_2, j)}}} \right\} \le \delta.$$

Summing over all criteria $j \in [k]$ along with a union bound, we obtain

$$\Pr \left\{ \|\mathbf{P}(\cdot, i_2) - \widehat{\mathbf{P}}(\cdot, i_2))\|_{\infty, 1} \ge ck \sqrt{\frac{\log(cdk/\delta)}{\min_{i_1, j} N_{(i_1, i_2, j)}}} \right\} \le \delta.$$

Finally, in order to obtain a bound on the maximum deviation in the $(\infty, 1)$-norm, we take a union bound over all $d$ choices of the index $i_2$, and apply inequality (16) to obtain

$$\max_{i_2} \|\mathbf{P}(\cdot, i_2) - \widehat{\mathbf{P}}(\cdot, i_2))\|_{\infty, 1} \le ck \sqrt{\frac{d^2 k}{n} \log\left( c \frac{dk}{\delta} \right)}$$

with probability exceeding $1 - \delta$. $\qquad\square$

A few comments regarding the corollary are in order. The above corollary suggests that the sample complexity required for obtaining an $\epsilon$-accurate solution with respect to the $\ell_1$ norm is $n = \widetilde{O}(\frac{d^2 k^3}{\epsilon^2})$. Observe that this bound is a factor of $k^2$ worse than the corresponding one for $\ell_\infty$ norm established in Corollary 1. This additional sample complexity occurs since for any vector $v \in \mathbb{R}^k$, we have $\|v\|_1 \le k\|v\|_\infty$. This implies that the error when measured with respect to $\ell_1$ can be upto $k$ times larger; since the sample complexity scales as $\frac{1}{\epsilon^2}$, the corresponding increase with respect to the number of criteria $k$ is quadratic.

## C.2 Optimization algorithms

Recall that Theorem 3 established that the objective function $v(\pi; \mathbf{P}, S, \|\cdot\|_q)$ is convex in $\pi$ and Lipschitz with respect to the $\ell_1$ norm. This implies that one could compute the plug-in solution $\widehat{\pi}_{\text{plug}}$ as a solution to a constrained optimization problem. In this section, we discuss a few specific algorithms based on zeroth-order and first-order methods for obtaining such a solution.

### C.2.1 Zeroth-order optimization

Zeroth-order methods for minimizing a function $f(x)$ over $x \in \mathcal{X}$ work with a function query oracle. That is, at each time step, the algorithm has access to an oracle which returns the value $f(x)$ for any point $x \in \mathcal{X}$. In our setup, since we are interested in minimizing the value function $v(\pi; \mathbf{P}, S, \rho)$ over $\pi \in \Delta_d$, such a function query requires access to the target set $S$ via an oracle $\mathcal{O}_S^0$ such that

$$\mathcal{O}_S^0(z) \to \min_{z_1 \in S} \rho(z, z_1) \,,$$

for the underlying distance function $\rho(\cdot)$. The oracle $\mathcal{O}_S^0$ essentially takes as input a score vector $z \in [0,1]^k$ and outputs the distance of this point to the target set $S$. Given this oracle, it is easy to see that for any $\pi$, one can compute the corresponding value function $v(\pi; \mathbf{P}, S, \rho)$.

There have been several algorithms proposed for optimization with such oracles when the underlying function $f$ is convex [23, 2, 48, 21, 38, 49] or non-convex, smooth [26]. The key idea in the proposed algorithms is to utilize the zeroth-order oracle to constuct estimates of the (sub-)gradient of the function $f$ using a class of techniques called *randomized smoothing*. The algorithms then differ in the construction of these estimates depending on the underlying randomness as well as on the number of oracle calls during each time step.

Given the results of Theorem 3, we can restrict our focus on algorithms for the class of convex Lipschitz function $f$. To this end, Shamir [49] proposed an algorithm for optimizing such functions which required *two* function evaluations at each time. The algorithm, adapted to the multi-criteria preference learning problem, is detailed in Algorithm 1. For our setup, we select the negative entropy regularization, $r(\pi) = \sum_i \pi_i \log(\pi_i)$ to suit the geometry of our domain $\mathcal{X} = \Delta_d$.

The proposed algorithm, maintains an estimate of the distribution, $\pi_t$, and at each time step $t$, queries the function value $v(\cdot; \mathbf{P}, S, \rho)$ at the following two points: $\pi_t + \delta u_t$ and $\pi_t - \delta u_t$, where $u$ is sampled uniformly from the Euclidean unit sphere and $\delta > 0$ represents the smoothing radius. Given these queries, the sub-gradient estimate, $\hat{g}_t$ is then obtained as:

$$\hat{g}_t := \frac{d}{2\delta} \left( v(\pi_t + \delta u_t; \mathbf{P}, S, \rho) - v(\pi_t - \delta u_t; \mathbf{P}, S, \rho) \right) u_t .$$

The sub-gradient estimate is then used to update the parameter estimate $\pi_{t+1}$ using the mirror descent algorithm with the specified regularization function. The zeroth-order method in Algorithm 1 does not require the underlying function to be smooth and hence works for our problem setup with arbitrary non-differentiable distance functions. We can now obtain the following convergence result, based on Theorem 1 from the work of Shamir [49].

**Proposition 5.** *Suppose the conditions of Theorem 3 hold, and that Algorithm 1 is run for $T$ iterations with step-size $\eta_t = \frac{c}{k^{1/q}\sqrt{dT}}$ and smoothing radius $\delta = \frac{c\log d}{\sqrt{T}}$, and produces a sequence $\pi_1, \pi_2, \ldots, \pi_T$. Then we have*

$$v\left(\bar{\pi}_T; \mathbf{P}, S, \|\cdot\|_q\right) \leq \min_{\pi \in \Delta_d} v(\pi; \mathbf{P}, S, \|\cdot\|_q) + ck^{\frac{1}{q}} \cdot \sqrt{\frac{d\log^2 d}{T}}$$

*where $\bar{\pi}_T = \frac{1}{T} \sum_{t=1}^T \pi_t$.*

*Proof.* By Theorem 3, the value function $v(\pi; \mathbf{P}, S, \|\cdot\|_q)$ is convex and $L_\mathsf{v} = k^{\frac{1}{q}}$-Lipschitz with respect to $\|\cdot\|_1$. Also, the choice of the regularizer $r(\pi) = \sum_i \pi_i \log(\pi_i)$ is 1-strongly convex with respect to the $\|\cdot\|_1$. Plugging in the above values in Theorem 1 from [49] establishes the above convergence rate. $\square$

Thus, in order to obtain a distribution $\hat{\pi}$ that is $\epsilon$-close to $\pi^*$ in function value, we need to run Algorithm 1 for $T = O\left(\frac{k^{\frac{2}{q}} d\log^2 d}{\epsilon^2}\right)$ iterations. Also, note that each iteration of the algorithm requires $d$ calls to the oracle $\mathcal{O}_S^0$. Therefore the total oracle complexity of the procedure is $O\left(\frac{k^{\frac{2}{q}} d^2 \log^2 d}{\epsilon^2}\right)$.

### C.3 First-order optimization

In this section, we look at first-order methods to compute the plug-in estimator. Let us denote by $\partial v(\pi)$ the set of sub-differentials of the function $v(\cdot; \mathbf{P}, S, \|\cdot\|)$ evaluated at $\pi$. Further, let the set $\Gamma(\pi)$ denote the set of maximizers for a policy $\pi$, that is,

$$\Gamma(\pi) = \left\{ \tilde{\pi} \in \Delta_d \mid \tilde{\pi} \in \operatorname*{argmax}_{\pi_2 \in \Delta_d} \min_{z \in S} \left[ \|\mathbf{P}(\pi, \pi_2) - z\| \right] \right\} . \tag{26}$$

Note that both of these quantities depend implicitly on the tuple $(S, \mathbf{P}, \|\cdot\|)$, but we have dropped this dependence in the notation. Given the setup above, Lemma 5 below characterizes this set $\partial v(\pi)$ for any smooth $\ell_q$ norm (with $1 < q < \infty$).

**Algorithm 2:** First-order method for multi-criteria preference learning

---

**Input:** Time steps $T$, step size $\eta$
**Initialize:** $\theta_1 = 1_k$
**for** $t = 1, \ldots, T$ **do**

  Set the distribution $\pi_t = \frac{\theta_t}{\|\theta_t\|_1}$

  Obtain $g_t \in \text{conv}\left\{ \frac{\mathbf{P}(\cdot, \pi_2)[\mathbf{P}(\pi_t, \pi_2) - \Pi_S(\mathbf{P}(\pi_t, \pi_2))]}{\|\mathbf{P}(\pi_t, \pi_2) - \Pi_S(\mathbf{P}(\pi_t, \pi_2))\|_q} \mid \pi_2 \in \Gamma(\pi_t) \right\}$   [See eq.(26) for $\Gamma(\pi_t)$]

  Update $\theta_{t+1,i} = \pi_{t,i} \exp(-\eta g_{t,i})$

**Output:** $\bar{\pi}_T = \frac{1}{T} \sum_{t=1}^{T} \pi_t$

---

**Lemma 5.** *Suppose that the distance is induced by a smooth $\ell_q$ norm for $1 < q < \infty$. Then the set of sub-differentials of $v$ at $\pi$ is given by:*

$$\partial v(\pi) = conv\left\{ \frac{\mathbf{P}(\cdot, \pi_2)\left[\mathbf{P}(\pi, \pi_2) - \Pi_S(\mathbf{P}(\pi, \pi_2))\right]}{\|\mathbf{P}(\pi, \pi_2) - \Pi_S(\mathbf{P}(\pi, \pi_2))\|_q} \mid \pi_2 \in \Gamma(\pi) \right\} \ ,$$

*where $\Pi_S(z)$ denotes the unique projection of the point $z$ onto set $S$ along $\|\cdot\|_q$.*

We defer the proof of the above lemma to later in the section. Note that in order to access such a sub-gradient, we need access to an oracle $\mathcal{O}_S^1$ that provides projection queries of the form

$$\mathcal{O}_S^1(z) \to \underset{z_1 \in S}{\text{argmin}}\, \rho(z, z_1).$$

The oracle $\mathcal{O}_S^1$ takes in a point $z$ and outputs the closest point in the set $S$ to this point. Given such an oracle, we can compute the sub-gradient of the function $v(\pi; \mathbf{P}, S, \rho)$ using Lemma 5 by evaluating it at the point given by $\mathbf{P}(\pi, \pi_2)$ for some $\pi_2 \in \Gamma(\pi)$.

Given access to such a projection oracle $\mathcal{O}_S^1$, we detail out a procedure based on a standard implementation of mirror descent with entropic regularization (or Exponentiated gradient method) in Algorithm 2 to minimize the objective $v(\pi; G)$. Note that we select the negative entropy function, $r(\pi) = \sum_i \pi_i \log(\pi_i)$, as the regularization function for the mirror descent procedure since our parameter space is given by the simplex $\Delta_k$ and the negative entropy function is known to be 1-strongly convex with respect to $\|\cdot\|_1$ over this space.

The algorithm works by maintaining at each time instance a distribution $\pi_t$ over the set of objects and updates it via an exponentiated gradient update. That is, the sub-gradient $g_t$ is evaluated at the current point $\pi_t$ using access to both $\mathcal{O}_S^1$ and $\mathcal{O}_S^0$, and is used to update each coordinate of the variable $\theta_t$. The updated distribution $\pi_{t+1}$ is obtained via a KL-projection of $\theta_t$ onto the simplex $\Delta_k$, which can be shown to be equivalent to the normalization $\theta/\|\theta\|_1$. We now proceed to prove a convergence result for this gradient-based Algorithm 2, based on a standard analysis of the mirror descent procedure (for example, see [14, Theorem 4.2]).

**Proposition 6.** *Suppose the conditions of Theorem 3 hold and consider any $\ell_q$-norm for $1 < q < \infty$. Suppose that running Algorithm 1 for $T$ iterations with step-size $\eta_t = \frac{1}{k^{1/q}}\sqrt{\frac{2\log d}{T}}$ produces a sequence $\pi_1, \pi_2, \ldots, \pi_T$. Then we have*

$$v(\bar{\pi}_T; \mathbf{P}, S, \|\cdot\|_q) \leq \min_{\pi \in \Delta_d} v(\pi; \mathbf{P}, S, \|\cdot\|_q) + k^{\frac{1}{q}} \cdot \sqrt{\frac{2\log d}{T}}$$

*where $\bar{\pi}_T = \frac{1}{T}\sum_{t=1}^{T} \pi_t$.*

*Proof.* Note that the function $v(\pi; \mathbf{P}, S, \|\cdot\|_q)$ is convex and $k^{\frac{1}{q}}$-Lipschitz with respect to the $\ell_1$ norm from Theorem 3. Further, the mirror map given by negative entropy function is 1-strongly convex with respect to $\|\cdot\|_1$. Plugging in these values in Theorem 4.2 from [14] establishes the required convergence rate. $\qquad\square$

In order to obtain an $\epsilon$-accurate solution in function value, it suffices to run the above algorithm for $T = O\left(\frac{k^{\frac{2}{q}}\log d}{\epsilon^2}\right)$ iterations, with each iteration using 1 call to the oracle $\mathcal{O}_S^1$ and $d$ calls

to the oracle $\mathcal{O}_S^0$ (to obtain the set $\Gamma$). Thus, we see that the total oracle complexity changes as $\mathcal{O}_S^1 : O\left(\frac{k^{\frac{2}{q}}\log d}{\epsilon^2}\right)$ calls and $\mathcal{O}_S^0 : O\left(\frac{k^{\frac{2}{q}}d\log d}{\epsilon^2}\right)$ calls – effectively, an $O(d\log d)$ decrease in the calls to $\mathcal{O}_S^0$ is compensated by a corresponding increase of $O(\frac{\log d}{\epsilon^2})$ calls to the stronger oracle $\mathcal{O}_S^1$.

**Proof of Lemma 5.** Consider the function $\phi(\pi_1, \pi_2) = \max_{z \in S}\|\mathbf{P}(\pi_1, \pi_2) - z\|$ over the domain $\pi_2 \in \Delta_d$. For any fixed $\pi_2$, we have that the function $\phi(\pi_1, \pi_2)$ is convex in $\pi_1$. Thus, by Danskin's theorem, we have that the subdifferential set is given by:

$$\partial v(\pi) = \text{conv}\left\{\frac{\partial \phi(\pi, \pi_2)}{\partial \pi} \mid \pi_2 \in \Gamma(\pi)\right\}, \tag{27}$$

where conv represents the convex hull of the set. Let us now focus on the partial derivative $\frac{\partial \phi(\pi, \pi_2)}{\partial \pi}$ for any $\pi_2$ which is a maximizer. This partial derivative involves differentiation of a metric projection onto a convex set, which has been studied extensively in the literature of convex analysis [41, 58, 5]. Recently, Balestro et al. [8] established that for distance functions given by smooth norms, the derivative of metric projection for any $z \notin S$ is given by:

$$\nabla\rho(z, S) = \nabla\min_{z_2 \in S}\|z - z_2\| = \frac{z - \Pi_S(z)}{\|z - \Pi_S(z)\|},$$

where $\Pi_S(z)$ denotes the unique projection of the point $z$ onto set $S$. Combining this with the chain rule of differentiation, we have that:

$$\frac{\partial \phi(\pi, \pi_2)}{\partial \pi} = \frac{\mathbf{P}(\cdot, \pi_2)\left[\mathbf{P}(\pi, \pi_2) - \Pi_S(\mathbf{P}(\pi, \pi_2))\right]}{\|\mathbf{P}(\pi, \pi_2) - \Pi_S(\mathbf{P}(\pi, \pi_2))\|_q}.$$

The above, in conjunction with equation (27) establishes the desired claim. □

# D  Details of user study

In this section, we provide the deferred details of the user study from Section 4.

**Self-driving environment.** The self-driving environment consists of an autonomous car which can be controlled by providing real-valued inputs acceleration and angular acceleration at every time step. We allow the policies to have access to the dynamics of this environment. Observe that there is no explicit reward function in the environment and each policy differs in the way it optimizes a chosen reward function to drive the car forward in a safe manner.

**Policies.** The MPC based Policies A-E were constructed by optimizing linear rewards comprising features F1-F9 as

- F1 Distance from the starting point along y-axis.
- F2 Velocity of the autonomous car.
- F3 Distance from the center of each lane.
- F4 Gaussian collision detector for nearby objects.
- F5 Collision detector which works at smaller radii than F4.
- F6 Over-speeding feature which penalizes higher speeds.
- F7 Reward for over-taking vehicles in the front.
- F8 Gaussian off-road detector.
- F9 Reward to promote speeding up near obstacles.

For each of the base policy, we set the weights of the features to encode different driving behaviors.

- Pol A programmed to prefer the right-most lane and progress forward at a slow speed.
- Pol B programmed to prefer the left-most lane and move forward as fast as possible.
- Pol C programmed to be conservative, avoids collision and proceeds forward.
- Pol D programmed to get attracted towards other cars and obstacles.
- Pol E programmed to prefer center lane and exhibit opportunistic behavior by moving ahead of other cars.

**Details of target set and linear weights.** We selected the two data-oblivious sets to trade-off between the criteria C1-C5 as

$$S_1 = \{z \mid z \in [0,1]^5, z_1 \geq 0.3, z_2 \geq 0.3, z_3 \geq 0.2, z_4 \geq 0.3, z_5 \geq 0.4\},$$
$$S_2 = \{z \mid z \in [0,1]^5, z_1 \geq 0.25, z_2 \geq 0.25, z_3 \geq 0.25, z_4 \geq 0.25, z_5 \geq 0.25, z_1 + z_5 \geq 0.9\}. \tag{28}$$

In addition, we selected 9 set of weights $w_{1:9}$ for linearly combining the different criteria.

$w_1$: Average of the users' self-reported weights.

$w_2$: Weight vector obtained by regressing the overall criterion on C1-C5 with squared loss as

$$w_2 \in \underset{w \in \Delta_5}{\text{argmin}} \sum_{i_1, i_2} (\mathbf{P}_{\mathsf{ov}}(i_1, i_2) - \sum_j w(j)\mathbf{P}^j(i_1, i_2))^2.$$

$w_3$: Weight obtained by regressing Bradley-Terry-Luce (BTL) scores. The BTL parametric model assumes a real-valued score $v_i$ for each policy and posits that $\Pr(\text{Pol } i \succeq \text{Pol } j) = \exp(v_i)/\exp(v_i) + \exp(v_j)$. Denoting the scores obtained from the overall preferences by $v^{\mathsf{ov}}$ and those obtained from the individual criteria by $v^j$ for $j \in [5]$, the weight

$$w_2 \in \underset{w \in \Delta_5}{\text{argmin}} \sum_i (v_i^{\mathsf{ov}} - \sum_j w(j)v_i^j)^2.$$

$w_4$: Data-oblivious weight $w_4 = [0.2, 0.2, 0.2, 0.2, 0.2]$.

$w_5$: Data-oblivious weight $w_5 = [0.25, 0.5/3, 0.5/3, 0.5/3, 0.25]$.

$w_6$: Data-oblivious weight $w_6 = [0.30, 0.4/3, 0.4/3, 0.4/3, 0.30]$.

$w_7$: Data-oblivious weight $w_7 = [0.5/3, 0.5/3, 0.25, 0.5/3, 0.25]$.

$w_8$: Data-oblivious weight $w_8 = [0.4/3, 0.4/3, 0.3, 0.4/3, 0.30]$.

$w_9$: Data-oblivious weight $w_9 = [0.3, 0.1/2, 0.3, 0.1/2, 0.3]$.

The set of data oblivious weights were chosen to account for different trade-offs along the criteria C1-C5 including the uniform weight $w_4$.

**Data Collection.** Table 1 shows the comparison data collected from the Mturk users in both the phases of the experiment. The entry $i, j$ of the comparison matrices represents the fraction of users which preferred Policy $i$ over Policy $j$. The top 5 rows and columns of each matrix correspond to the baseline policies while the bottom rows correspond to the two randomized policies R1 and R2 obtained as the Blackwell winner corresponding to sets $S_1$ and $S_2$ respectively.

In addition, we would like to highlight some details from an experiment design perspective. Since the experiment was run in two phases, we could not guarantee the same set of subjects to participate in both parts of the experiment. In order to limit distribution shifts, we restricted the nationality of the subjects to United States and began both the phases on the same time and day of the week. Also, in order to prevent biased evaluations, the ordering of the policy pairs as well as the ordering policies within a comparison was randomized across the users.

Figures 3, 4 and 5 shows the experiment setup we used for obtaining comparison data from Amazon Mechanical Turk users consisting of the instructions, the policy comparison page and the questionnaire that the users were asked to fill out.

**Implementation Details.** The computation of the Blackwell winner for the different target set $S$ was done using the CVX package in Matlab. For the MPC policies, the horizon length $H$ was set to be 18, three times the planning horizon $= 6$.

|     | A | B | C | D | E |
|-----|------|------|------|------|------|
| A | 0.50 | 0.64 | 0.45 | 0.41 | 0.39 |
| B | 0.36 | 0.50 | 0.30 | 0.30 | 0.25 |
| C | 0.55 | 0.70 | 0.50 | 0.55 | 0.57 |
| D | 0.59 | 0.70 | 0.45 | 0.50 | 0.52 |
| E | 0.61 | 0.75 | 0.43 | 0.48 | 0.50 |
| R1 | 0.49 | 0.80 | 0.22 | 0.46 | 0.29 |
| R2 | 0.49 | 0.88 | 0.66 | 0.61 | 0.41 |

(a) C1: Aggressiveness

|     | A | B | C | D | E |
|-----|------|------|------|------|------|
| A | 0.50 | 0.57 | 0.50 | 0.50 | 0.41 |
| B | 0.43 | 0.50 | 0.30 | 0.39 | 0.45 |
| C | 0.50 | 0.70 | 0.50 | 0.43 | 0.59 |
| D | 0.50 | 0.61 | 0.57 | 0.50 | 0.57 |
| E | 0.59 | 0.55 | 0.41 | 0.43 | 0.50 |
| R1 | 0.46 | 0.71 | 0.32 | 0.51 | 0.39 |
| R2 | 0.51 | 0.71 | 0.61 | 0.59 | 0.51 |

(b) C2: Predictability

|     | A | B | C | D | E |
|-----|------|------|------|------|------|
| A | 0.50 | 0.16 | 0.25 | 0.32 | 0.30 |
| B | 0.84 | 0.50 | 0.89 | 0.82 | 0.68 |
| C | 0.75 | 0.11 | 0.50 | 0.73 | 0.61 |
| D | 0.68 | 0.18 | 0.27 | 0.50 | 0.41 |
| E | 0.70 | 0.32 | 0.39 | 0.59 | 0.50 |
| R1 | 0.73 | 0.22 | 0.76 | 0.78 | 0.76 |
| R2 | 0.90 | 0.24 | 0.44 | 0.66 | 0.66 |

(c) C3: Quickness

|     | A | B | C | D | E |
|-----|------|------|------|------|------|
| A | 0.50 | 0.59 | 0.45 | 0.57 | 0.39 |
| B | 0.41 | 0.50 | 0.32 | 0.34 | 0.32 |
| C | 0.55 | 0.68 | 0.50 | 0.48 | 0.59 |
| D | 0.43 | 0.66 | 0.52 | 0.50 | 0.50 |
| E | 0.61 | 0.68 | 0.41 | 0.50 | 0.50 |
| R1 | 0.44 | 0.80 | 0.20 | 0.39 | 0.24 |
| R2 | 0.41 | 0.80 | 0.71 | 0.59 | 0.39 |

(d) C4: Conservativeness

|     | A | B | C | D | E |
|-----|------|------|------|------|------|
| A | 0.50 | 0.52 | 0.41 | 0.50 | 0.43 |
| B | 0.48 | 0.50 | 0.32 | 0.55 | 0.55 |
| C | 0.59 | 0.68 | 0.50 | 0.55 | 0.57 |
| D | 0.50 | 0.45 | 0.45 | 0.50 | 0.50 |
| E | 0.57 | 0.45 | 0.43 | 0.50 | 0.50 |
| R1 | 0.54 | 0.68 | 0.32 | 0.49 | 0.41 |
| R2 | 0.63 | 0.73 | 0.59 | 0.61 | 0.54 |

(e) C5: Collision Risk

|     | A | B | C | D | E |
|-----|------|------|------|------|------|
| A | 0.50 | 0.39 | 0.25 | 0.43 | 0.34 |
| B | 0.61 | 0.50 | 0.30 | 0.50 | 0.50 |
| C | 0.75 | 0.70 | 0.50 | 0.57 | 0.61 |
| D | 0.57 | 0.50 | 0.43 | 0.50 | 0.48 |
| E | 0.66 | 0.50 | 0.39 | 0.52 | 0.50 |
| R1 | 0.66 | 0.76 | 0.29 | 0.59 | 0.39 |
| R2 | 0.66 | 0.73 | **0.66** | 0.56 | 0.51 |

(f) Overall Preferences

**Table 1.** Each matrix consists of pairwise comparisons between policies elicited from a user study with around 50 participants on Mturk. An entry $i, j$ of the comparison matrices represents the fraction of users which preferred Policy i over Policy j. Policies A-E comprise the base set of policies while Policies R1-R2 are the randomized Blackwell winners obtained from the sets in equation (28). While Policy C is the overall von Neumann winner, Policy R2 is preferred over it by 66% of the users.

# Instructions

In this experiment, the objective is to select amongst a given alternatives of self-driving cars based off on their performance along different objectives.

We will show you self-driving cars, operated by different softwares (or algorithms) which leads them to exhibit different behaviors in different environments. In each part of the experiment, we will show you a pair of self-driving softwares and how they behave in certain environments. The behavior of the driving policies will be shown from a bird's eye view.

We will then ask you comparative questions which will ask you to select one of the driving softwares according to a specified criterion and ask you the reasoning behind your choices.

I understand ➜

# Instructions

**During the experiment, please remember the following:**

- It is important that you carefully observe the behavior of the softwares in the provided environments before responding to the following questions based on that.
- You will be allowed to proceed to the next part of the experiment only once you have responded to **all the comparison questions** and have specified the appropriate justification for your choices.
- Please note that the main car driven by the software will be coloured in **Orange** while the other companion cars will be shown in **Black**.
- Each of the softwares has been labelled as Software {G, H}. Across the different experiments, the naming of the software remains consistent. For instance, Software A will remain the same software during each of the individual experiments of the survey. Note that some of these policies make use of randomization and their behavior might differ across experiments.

← Previous          I understand ➜

**Figure 3.** Instructions provided to the users before the experiment began. The users were asked to compare behavior of policies and were told to expect some policies to exhibit a randomized behavior.

# Experiment Progress: 1/1

### (Please allow all 6 images to load below before responding to the questions)

(A convenient way to proceed would be to compare the behavior of the two softwares across each of the environments on the top and bottom row, that is, first along environment 1, 2 and then 3.)

## Self Driving Software H

Environment 1&emsp;&emsp;&emsp;&emsp;Environment 2&emsp;&emsp;&emsp;&emsp;Environment 3

## Self Driving Software G

Environment 1&emsp;&emsp;&emsp;&emsp;Environment 2&emsp;&emsp;&emsp;&emsp;Environment 3

**Figure 4.** Layout of the experiment where each panel shows a GIF exhibiting a Policy controlling the autonomous vehicle in one of the worlds of the environment. The users were instructed to compare behaviors across each of the columns before proceeding to answer the questions.

# Questions

Q1*. Which of the two softwares exhibits a less aggressive behavior? :

   ○ Software H      ○ Software G

Q2*. Which of the two softwares is more predictable in their behavior? That is, for which of the two softwares do you think you will be able to anticipate its performance in a new environment. :

   ○ Software H      ○ Software G

Q3*. Which of the two softwares will get you to your destination the quickest?:

   ○ Software H      ○ Software G

Q4*. Which of the two softwares is more conservative in its driving approach?:

   ○ Software H      ○ Software G

Q5*. Which of the two softwares if has a lower risk of collision with another car or an obstacle?:

   ○ Software H      ○ Software G

Q6*. [Overall Preference] Imagine you were to select one of the two softwares to get you to your destination. Which of the two softwares would you prefer?:

   ○ Software H      ○ Software G

* Please provide a brief sentence about how you made your selections: (Press 'Enter' after typing the sentence)

[                                                                                              ]

* For each of the following characteristic, please indicate their relevance in determining the overall preference betweem the softwares. Please take into account all the expriments that you completed in this study. *(5 = extremely important, 1 = had little importance)*

Aggresiveness of the software:
○ 1      ○ 2      ○ 3      ○ 4      ○ 5

Predictability of the software:
○ 1      ○ 2      ○ 3      ○ 4      ○ 5

Speed or quickness of the software:
○ 1      ○ 2      ○ 3      ○ 4      ○ 5

Conservativeness of the software:
○ 1      ○ 2      ○ 3      ○ 4      ○ 5

Collision Risk of the software:
○ 1      ○ 2      ○ 3      ○ 4      ○ 5

**Figure 5.** Layout of the questions panel comprising the 6 comparison questions and the form for reporting the relevance of each criterion in the overall evaluation.