[Reviews · NeurIPS 2020]

Review 1

Summary and Contributions: The paper focuses on learning user preferences in multiple criteria setting. First, the paper introduces the concept of Blackwell winner taking inspiration from Blackwell approachability. The Blackwell winner generalizes many known notions of wining solution concepts. Important aspect of the paper is a way of non-linear aggregation of preferences across multiple criteria, where the Blackwell winner can be computed by optimizing a convex optimization problem. Theoretically, the paper shows, in the passive setting, that a “plug-in” estimator achieves (near-)optimal minimax sample complexity. Lastly, the authors elaborate on a user-study on autonomous driving, which shows the efficacy of the proposed method.

Strengths: 1. The paper's approach is elegant given the complex nature of multi-criteria problem. 2. Conducting a user-study is also one of the good aspects of the paper. 3. The paper is well-written and easy to understand (some things can be clarified though).

Weaknesses: 1. I believe the paper should have also focused on the algorithmic aspects of the solution. Once the concept of Blackwell winner is proposed, the novelty of the paper seems limited. 2. Some more details on the user study would have made the empirical claims stronger. 3. Minor: One thing that is problematic in the approach is the selection of target sets. I hope something can be done about it. Nevertheless, it is good that the authors acknowledge this limitation.

Correctness: The theoretical claims in the paper seem sound and valid. Empirical methodology is also good; however, some more details would have helped.

Clarity: Yes, the paper is well written.

Relation to Prior Work: The paper also does fairly good job of explaining the prior work and mentioning the differences.

Reproducibility: Yes

Additional Feedback: Some of the major/minor concerns in chronological order are as follows: 1. Please mention the citation for Condorset winner and Borda winner when you first mention them in the introduction. 2. It would be interesting to see how to construct active approaches for the given problem setting. The authors acknowledge this aspect, anyway in the paper. 3. The theoretical results are shown for a specific norm and specific type of target sets; however, the definitions make use of general norms and target sets. In my opinion, this makes the results fairly restrictive. 4. Line 265 -- should mention "asymptotically" in the end. 5. After the authors propose Blackwell winner, the paper's novelty seem to be fairly limited. The analysis job is well-done. However, the authors should have shed more light on the algorithmic developments. Just from writing perspective, too, adding algorithms to the main paper will help the reader. Maybe the authors can combine Proposition 2 and Theorem 2 to just save space for algorithms. 6. Overall the approach seems fine; however, the choice of target set seems problematic because it directly affects the elicitation procedure. I am wondering if there is a way to elicit the target set as well. 7. line 295 -- Why only H = 18 was used? Was it based on trial and error? Was it allowing for fair comparisons? 8. I believe the queries that were asked to the users were complicated. Perhaps that's why the user study is small. I would encourage authors to focus more on reducing the complexity of the query and also on active approaches. 9. Overall conducting a user-study is a good aspect of the paper. However, some details for the user study are missing such as the kind of tests performed and the sample sizes. For example, in line 329 only averages are mentioned, standard deviation is missing. Also it would have been better if there was some statistical test conducted to know that the weights are actually not a random consequence. Also, I am not sure how the oblivious weights are chosen. 10. It is good to see that the authors have discussed limitation of their approaches. Whatever possible limitation I could think of was already acknowledged by the authors in future work or broader aspects as limitations. Overall, the paper is good. Only that I believe the novelty is a bit limited with very less algorithmic development. Other than that, the changes that I have suggested can be done easily if the paper is accepted. ----- After Author's response ---------- I thank the authors for providing responses to the questions. One of my concerns was limited novelty, to which now I think I was little less supportive earlier. After reading the authors responses, I am more positive about the novelty. Also, authors have clarified that the upper bounds in Theorem 1 are for general `q norms and general target sets, and only the lower bounds focus on the \infty norm. So that's also a positive aspect which I might have missed earlier. Hence, I would like to increase my score to 7 for this paper.


Review 2

Summary and Contributions: The paper proposes a technique for combining pairwise comparisons along multiple criteria into an overall winner. The technique is based on the game-theoretic notion of a Blackwell winner, but is adapted here. The paper provides a thorough theoretical dsicussion, the proof of sample complexity bounds, and an evaluation in a small empirical study based on a mechanical turk survey.

Strengths: - interesting problem - thorough theoretical analysis - excellent presentation

Weaknesses: - empirical evaluation on a single small domain is maybe a bit limited - preference dimensions of the evaluation are not quite clear

Correctness: The key results are theoretical proofs, which appear to be correct. The experimental study in a single domain is maybe a bit limited, and somewhat unclear in its design. E.g., the 5 preference criteria seem to be quite correlated (e.g., C1: less agressive and C5: less collision risk) or inversely correlated (e.g., C3: more quick and C4: more conservative). I think a study involving a more diverse set of preference criteria (such as color, size, energy consumption, power in a car purchase) would have been more convincing (but probably also less likely to yield good results). I would also think that there should be benchmark datasets for such tasks available (but cannot make a suggestion myself).

Clarity: Yes, very much so. The paper strikes a very good balance between formal rigor and intuitive explanations, it is a pleasure to read.

Relation to Prior Work: The paper has a discussion of related work in neighboring fields such as game theory, MCDA, and computational social choice, but it completely misses out on a discussion of related work in machine learning, where learning from pairwise comparisons has also received quite some attention (e.g., Eyke Hüllermeier, Johannes Fürnkranz, Weiwei Cheng, Klaus Brinker: Label ranking by learning pairwise preferences. Artif. Intell. 172(16-17): 1897-1916 (2008))

Reproducibility: Yes

Additional Feedback: The paper does not say whether the dataset is available, which would, of course, be desirable.


Review 3

Summary and Contributions: This paper studies the problem of generating a candidate by learning the preferences of multiple agents, using a motivating example from RL policies in autonomous driving. The main contribution is in extending the problem studied in prior literature on von Neumann winner, to multiple criteria structures behind preference profiles. This is done by adapting the concept of Blackwell approachability and target sets. The proposed model is shown to generalize the Von-Neumann winner model, and is shown to reduce to the latter when there is only one criterion. The authors show an upper bound on estimation error, and an information-theoretic lower bound for a plug-in estimator based on empirical preference estimates. The plug-in estimator is shown to be computable using convex programming. Finally, the authors provide a user study using crowdsourced preferences.

Strengths: The applicability to preference learning under multiple criteria is compelling, and the error and sample complexity guarantees are analytically helpful. This idea might have broader applications beyond the specific example the authors use. This paper also provides theoretical guarantees showcasing the learnability of the blackwell winner along with a user study showing its applicability in real world scenarios.

Weaknesses: The user study is limited in scope and scale, which limits one’s ability to deduce how users will act in other domains. The authors do not discuss how one might choose the target set S on which their solution concept is critically dependent. This becomes important since, in their user study, the choice of S is detrimental to their solution being better than the von Neumann winner. It seems that even the problem of determining what S might be is a challenge. For the error bound results, the authors assume a certain underlying distribution (uniform random) from which the data is sampled which is very different from traditional learning theory (which assumes nothing about the underlying distribution) and provide no justification for it. This naturally limits applicability. Furthermore, this seems to suggest that the users have some amount of control in choosing data points while creating the dataset in which case it is a little strange that the authors don’t assume an oracle which gives users exactly what data they want especially since this is what they do in their study. The algorithm to compute the Blackwell winner requires access to an oracle which outputs the distance between a point and a set. There is not much that is said about this oracle and its computational tractability. While for a few norm functions (like the L-infinity norm), this can be done in polynomial time using a constrained optimisation program (linear or quadratic), it is unclear whether such an oracle would exist for all the commonly used norm functions. Furthermore, even in the case where the oracle is computable, a single run of the oracle may itself be very expensive computationally, severely affecting the scalability of the algorithm.

Correctness: Yes

Clarity: Although it would help to define \rho in Appendix section A line 521 for readers who want to read about Blackwell approachability before they read about the Blackwell winner. Typos: Line 39: …example, the... Line 346: ...preferred Policy C to R2… Line 737: There is no equation labelled (iii) but I think this is referring to the equation right before Line 736

Relation to Prior Work: Yes

Reproducibility: Yes

Additional Feedback: I have two questions: Are the numbers in Fig 1(b) based on real data? What method was used to compute the Blackwell winner in the user study section? In particular, what algorithm was used for the Oracle?


Review 4

Summary and Contributions: ###########################Post-Rebuttal############################ In the rebuttal, the authors provide a brief comparison between the Pareto-optimal Set and their work. It is good. But I still do not believe that finding a predefined linear combination of criteria could mitigate the conflict across criteria. Moreover, there is no general guidance on how to choose the weights. There are no extra experiments to validate the theories. I have to keep my previous in the post-rebuttal phase. ################################################################## 1. This paper proposes a multi-criteria framework for preference aggregation from pairwise comparison possibilities. 2. The core of this framework lies the notion of the black-well winner, which is a multi-objective extension of the von Neumann winner. 3. Guided by the Blackwell’s definition of the target set, the authors provide relaxation of the black-well winner by considering the preference distributions that minimize the maximal distance to the target set. 4. Theoretical analyses show that the generated framework naturally embraces two special cases: (a) single-criteria aggregation and (b) linear combinations of multi-criteria scores. Moreover, it also includes much more complicated settings where the scores from multi-criteria are aggregated in a non-linear manner. Finally, the authors also provide lower and upper error bounds for computing the Blackwell winner from samples of pairwise comparisons along the various criteria.

Strengths: 1. This paper is well-written with clear logic. I feel comfortable to follow the main idea of this paper. 2. This paper studies an interesting problem where preference aggregation is carried out with multiple objectives. This is certainly a more realistic setting than the traditional single-objective studies, which is not fully understood by most of the prior studies. 3. The main idea of this paper is well-supported by clarified definitions and rigorous theoretical analysis

Weaknesses: 1.While the paper enjoys some strengths, my major concern is that the way the conflict across different objectives are handled with. In this paper, the conflict s are handled with the so-called target-set, which is predefined in most cases. This makes the whole method ad-hoc and restricts further applications. In terms of multi-objective optimization, the conflicts are often solve by employing the Pareto optimal solutions. It is well-known that one could use gradient-based methods in most cases to reach Pareto optimal solutions with high precisions. Consequently, I would like to see a comparison with these kinds of methods. 2. In some parts of the paper, the authors only give adopted methods without giving specific reasons. For example, on Page 5, [181]: The reason for using the distance function “|·|” is not given, which will make readers confused. 3. No reason is given for the choice of weights and parameters of target sets, and it is not stated whether they are sensitive to parameters and whether they are given artificially. 4. The experiment employs few datasets and baselines, which makes the advantage of the proposed method not convincing. The number of policies in a user study about autonomous driving looks a bit small. Considering these weaknesses, especially the ad-hoc target set, I think this paper is slightly below the standard NeurIPS.

Correctness: The theory, methodology are mostly correct.

Clarity: The paper is well-written and I have no difficulty to follow its main idea.

Relation to Prior Work: The discussion with the pareto-optimality-based MOO methods, which is also closely related with this work, is missing.

Reproducibility: Yes

Additional Feedback: 1. Why do the authors choose the distance function instead of other norms? Does it have better performances than other norms, or is it just chosen randomly? 2. In the experimental part of the appendix, the authors give many parameters. How are the weights and set parameters of this experiment selected?

[Author Response · NeurIPS 2020]

We thank the reviewers for their thoughtful feedback. We are pleased that the reviewers found the multi-criteria preference learning (MCPL) formulation interesting and the theoretical analysis insightful. We begin by clarifying some common concerns, including how the the user study is more nuanced than it seems, eliciting target sets (and how even pre-specified ones are useful, as our study shows), and why the passive sampling model is a natural first step towards an understanding of our framework. We would also like to highlight to R4 that our game-theoretic approach to the MCPL problem provides a systematic way to select amongst Pareto-optimal alternatives by using target sets.

**Scope of user study** [R1, R2, R3, R4]. While the reviewers appreciated the inclusion of a real-world user study, they raised some valid concerns regarding its scope and design. While we agree that the study with 5 base policies, 5 criteria and 50 participants is a small-scale study, we would like to highlight that it was already complex enough to capture a wide variety of behaviors: for example, a strict preference ordering in the overall comparison, as well as circularity in preferences along criterion C4 (conservativeness). Moreover, in spite of the limited sample size, our statistical tests demonstrate that our hypotheses are indeed supported within a $95\%$ confidence interval. Lines 912-917 further highlight some of the challenges that we faced in designing such a user study along with our adopted solutions. As mentioned by R2, we do not know of any existing benchmarks to evaluate MCPL methodologies. Our user study is a first step towards establishing a systematic benchmark for this domain and we shall make our anonymized data publicly available.

**Choice of target sets** $S$ [R1, R3]. Our framework currently puts the onus of the selection of the target set $S$ on the designer. This is applicable for a range of tasks like medical diagnosis wherein domain experts are required to assess these multi-criteria trade-offs (as we did in our user study). However, we agree with R1, R3 that several applications would benefit from a user-elicited target set. Classically, such elicitations have been studied in the economics literature and as stated in our Discussions section, studying such mechanisms is an important future direction.

**Passive sampling model** [R1, R3]. Our statistical analysis focuses on the passive sampling framework wherein each query is sampled uniformly at random. This framework captures several scenarios (including our self-driving case-study) wherein each user completes a questionnaire comprising *all* the comparison queries. In contrast, an active framework is useful for scenarios where some queries are easier to obtain, and understanding it from a theoretical perspective would require techniques (e.g from the Bandits literature), which is an interesting direction for future work.

**Reviewer R1.** – (**Novelty**) Our work formalizes the MCPL framework, proposes a game-theoretic solution concept (called the Blackwell winner) and provides a thorough theoretical analysis to understand the statistical and computational properties of this winner; the framework and the analysis together comprise our novel contributions. We will take R1's advice and add more algorithmic content from the appendix into Section 3.3.

– (**Theoretical Results**) Note that our upper bounds in Theorem 1 are for general $\ell_q$ norms and general target sets. While our lower bounds focus on the $\ell_\infty$ norm, Proposition 2 establishes an asymptotic lower bound for a wide class of target sets. In addition, we believe that our lower bound constructions are quite informative and are likely to be of independent interest, for instance, in understanding the stability of the Nash equilibrium to sampling errors.

– (**Details**) (i) We chose the horizon $H = 18$ to be three times the planning horizon $= 6$ for our MPC policies, allowing them to exhibit a varied behaviour over the complete trajectory without affecting their planning capabilities. (ii) The std dev for the weights were quite small (order 1e-3). (iii) The Condorcet and Borda winners have been cited in Line 29.

**Reviewer R2.** We thank R2 for their encouraging remarks. We shall include the missing references in the updated draft.

**Reviewer R3.** – (**General learning distributions**) Our sampling framework, which considers uniform distribution over the entries, can be extended to *any* distribution with complete support. The statistical guarantees will worsen by a condition number factor $\kappa = {}^{\text{max. prob}}/_{\text{min. prob}}$ where max (min) prob is the largest (smallest) probability of sampling.

– (**Access to target set**) Our algorithmic framework requires access to a pre-specified target set. One way to accomplish this is to write these target sets as an intersection of a finite number of half-spaces that in turn capture different trade-offs between criteria. For general convex sets, the distance oracle may be computationally intensive.

– (**Details**) (i) The numbers in Figure 1 are not from an actual study but are illustrative of results from our MTurk user-study; we shall clarify this. (ii) We used cvx optimization package in Matlab to compute the Blackwell winner.

**Reviewer R4.** – (**Comparison with Pareto-optimal solutions**) We agree that this is an important comparison. As a solution concept for multi-criteria preference learning, the Pareto-optimal set is insufficient since it does not provide a mechanism for selecting amongst those alternatives. Our framework with target sets provides a systematic way to specify preferences amongst these various Pareto optimal solutions. Further, if such preferences are indifferent between the Pareto-optimal solutions, the set of Blackwell winners will coincide with the Pareto-optimal set: in particular, one can show that for any preference tensor $\mathbf{P}$, there exists a target set $S$ such that the Blackwell winners for $(\mathbf{P}, S, \|\cdot\|_\infty)$ can recover the complete Pareto-optimal set. We will add a discussion comparing our approach with those for finding Pareto-optimal sets. In addition, we would like to point that our idea of using target sets as a selection mechanism is not arbitrary; it arises as a natural extension from real to vector-valued games, dating back to Blackwell's work.

– (**Distance metrics**) Since our example focuses on the $k = 1$ setup, all $\ell_q$ norms are equivalent to the absolute distance function $|\cdot|$. In comparison, our upper bounds in Theorem 1 concern general $\ell_q$ norms.

– (**User study parameters**) We tried to extensively cover different trade-offs through our choice of linearization weights and target sets. We shall add an ablation study on the robustness of the solutions to variations in weights and target sets.

[Meta-Review · NeurIPS 2020]

The paper studied preference aggregation via pairwise comparisons along multiple criteria. All reviewers find the problem setup interesting and appreciate the theoretical contribution novelty. I also share this sentiment, and find the paper a pleasure to read. The authors are strongly encouraged to take into account the reviews, in particular, to further strengthen the empirical analysis and discussions if possible, when preparing a revision.